# Cortex-wide BOLD fMRI activity reflects locally-recorded slow oscillation-associated calcium waves

Miriam Schwalm[1,2†], Florian Schmid[3†], Lydia Wachsmuth[3†], Hendrik Backhaus[1], Andrea Kronfeld[1], Felipe Aedo Jury[1], Pierre-Hugues Prouvot[1], Consuelo Fois[1], Franziska Albers[3], Timo van Alst[3], Cornelius Faber[3†], Albrecht Stroh[1†*]

[1]Focus Program Translational Neurosciences, Institute for Microscopic Anatomy and Neurobiology, Johannes Gutenberg-University Mainz, Mainz, Germany; [2]GRADE Brain, Goethe Graduate Academy, Goethe University Frankfurt am Main, Frankfurt, Germany; [3]Department of Clinical Radiology, University Hospital Münster, Münster, Germany

**Abstract** Spontaneous slow oscillation-associated slow wave activity represents an internally generated state which is characterized by alternations of network quiescence and stereotypical episodes of neuronal activity - slow wave events. However, it remains unclear which macroscopic signal is related to these active periods of the slow wave rhythm. We used optic fiber-based calcium recordings of local neural populations in cortex and thalamus to detect neurophysiologically defined slow calcium waves in isoflurane anesthetized rats. The individual slow wave events were used for an event-related analysis of simultaneously acquired whole-brain BOLD fMRI. We identified BOLD responses directly related to onsets of slow calcium waves, revealing a cortex-wide BOLD correlate: the entire cortex was engaged in this specific type of slow wave activity. These findings demonstrate a direct relation of defined neurophysiological events to a specific BOLD activity pattern and were confirmed for ongoing slow wave activity by independent component and seed-based analyses.

DOI: https://doi.org/10.7554/eLife.27602.001

*For correspondence: albrecht.stroh@unimedizin-mainz.de

†These authors contributed equally to this work

Competing interests: The authors declare that no competing interests exist.

## Introduction

Slow oscillation-associated slow wave activity is characterized by a waxing and waning of spontaneous neuronal firing arising from neuronal populations in deeper cortical layers (*Chauvette et al., 2010*; *Sanchez-Vives and McCormick, 2000*; *Stroh et al., 2013*), influencing neuronal excitability as well as stimulus-response properties of neuronal networks throughout the brain (*McGinley et al., 2015b*; *Petersen et al., 2003*; *Steriade et al., 1993c*; *1993b*). The slow wave rhythm as a 'default mode' of cortical network activity (*Sanchez-Vives and Mattia, 2014*) stands in contrast to the rather desynchronized, persistent state during rapid-eye-movement (REM) sleep and active wakefulness, dominated by low amplitude and high frequency cortical activity (*Constantinople and Bruno, 2011*; *McGinley et al., 2015a*; *Steriade et al., 2001*). A similar persistently active neuronal population activity can also be maintained by sedation (*Constantinople and Bruno, 2011*).

During slow wave activity, the corresponding low frequency (<1 Hz) component of field potential recordings reflects bimodality: *active* phases, in which cells are depolarized and *silent* periods with rather hyperpolarized membrane potentials (*Timofeev et al., 2001*), likely influenced through neuro-modulatory pathways (*Eggermann et al., 2014*). This oscillation is highly similar across cortical regions (*Ruiz-Mejias et al., 2011*), and also across species, from lizards to humans (*Achermann and Borbély, 1997*; *Buzsáki et al., 2013*; *Eschenko et al., 2006*; *Mölle et al., 2002*; *Shein-*

**eLife digest** When a person is in a deep non-dreaming sleep, neurons in their brain alternate slowly between periods of silence and periods of activity. This gives rise to low-frequency brain rhythms called slow waves, which are thought to help stabilize memories. Slow wave activity can be detected on multiple scales, from the pattern of electrical impulses sent by an individual neuron to the collective activity of the brain's entire outer layer, the cortex. But does slow wave activity in an individual group of neurons in the cortex affect the activity of the rest of the brain?

To find out, Schwalm, Schmid, Wachsmuth et al. took advantage of the fact that slow waves also occur under general anesthesia, and placed anesthetized rats inside miniature whole-brain scanners. A small region of cortex in each rat had been injected with a dye that fluoresces whenever the neurons in that region are active. An optical fiber was lowered into the rat's brain to transmit the fluorescence signals to a computer. Monitoring these signals while the animals lay inside the scanner revealed that slow-wave activity in any one group of cortical neurons was accompanied by slow-wave activity across the cortex as a whole. This relationship was seen only for slow waves, and not for other brain rhythms.

Slow waves seem to occur in all species of animal with a backbone, and in both healthy and diseased brains. While it is not possible to inject fluorescent dyes into the human brain, it is possible to monitor neuronal activity using electrodes. Comparing local electrode recordings with measures of whole-brain activity from scanners could thus allow similar experiments to be performed in people. There is growing evidence – from animal models and from studies of patients – that slow waves may be altered in Alzheimer's disease. Further work is required to determine whether detecting these changes could help diagnose disease at earlier stages, and whether reversing them may have therapeutic potential.

DOI: https://doi.org/10.7554/eLife.27602.002

*Idelson et al., 2016*; *Steriade et al., 2001*), suggesting well-defined conserved roles (*Destexhe et al., 2007*).

It has been shown that slow waves are propagating, most of the times in anterior-posterior direction over the cortical surface, eventually recruiting large cortical areas (*Busche et al., 2015*; *Massimini et al., 2004*; *Stroh et al., 2013*), resembling a travelling wave, likely mediated by recurrent excitatory activity (*Luczak et al., 2007*; *Massimini et al., 2004*; *Sanchez-Vives et al., 2017*; *Stroh et al., 2013*). In addition, slow wave activity – correlated to cortical slow wave events - can also be found in sub-cortical regions such as the thalamus (*Stroh et al., 2013*) and the hippocampus (*Busche et al., 2015*; *Hahn et al., 2006*), probably mediated by long-range excitatory connections (*Leong et al., 2016*). However, it remains to be established whether slow wave dynamics monitored in a local neural population are correlated with global functional network dynamics captured by whole-brain measures.

Slow wave activity of a spatially confined neural population can be detected by optic fiber-based calcium recordings (*Adelsberger et al., 2014*; *Grienberger et al., 2012*; *Stroh et al., 2013*). Upon staining of cells with fluorescent calcium indicators such as the synthetic dye Oregon Green BAPTA-1 (OGB-1; *Stosiek et al., 2003*) or by transduction of neurons with the genetically encoded GCaMP6 (*Chen et al., 2013a*), these optical recordings exclusively monitor activity of stained or transfected cells. Besides neurons, OGB-1 also stains astrocytes (*Garaschuk, 2013*), yet population-based recordings are dominated by neuronal action potential related calcium influx (*Grienberger et al., 2012*; *Grienberger and Konnerth, 2012*). Calcium transients acquired by the means of an optic fiber result from synchronous spiking of at least 30 cells at the recording site (*Grienberger et al., 2012*) and represent the integrated signals obtained mainly from neurons and the surrounding neuropil, as demonstrated by simultaneous two-photon calcium imaging (*Grienberger et al., 2012*; *Schulz et al., 2012*). Consequently, optic fiber-based calcium recordings are well suited for the recording of local slow wave activity.

BOLD fMRI provides a measure of brain-wide hemodynamic signals. The BOLD signal reflects changes in blood oxygenation, regional cerebral blood flow, and regional cerebral blood volume, linking these hemodynamic parameters to neuronal activity by neurovascular coupling. Resting state

fMRI (*Biswal et al., 1995*; *Fox and Raichle, 2007*; *Hutchison and Everling, 2014*) is increasingly employed to study large-scale correlations of brain activity fluctuations in low frequency bands. Neuronal correlates of spontaneous fluctuations in the BOLD signal have been identified by combining fMRI and LFP recordings at rest, subsequently correlating these two signals (*Chang et al., 2013*; *Hsu et al., 2016*; *Magri et al., 2012*; *Pan et al., 2013*; *Shmuel and Leopold, 2008*; *Thompson et al., 2014*). However, those studies used resting state fMRI methods to spatially map signal fluctuations which correlate to infra-slow fluctuations of the LFP signal, but they did not detect fMRI signal responses directly related to individual slow wave events.

Others correlated hemodynamic fluctuations with wide-field calcium imaging of excitatory neurons, convolving these two signals (*Ma et al., 2016b*) or compared functional connectivity during different, temporally transient patterns of calcium activity (*Matsui et al., 2016*). While these results support the notion of resting-state hemodynamics being coupled to underlying patterns of excitatory neuronal activity, particularly lower-frequency hemodynamic fluctuations were not well-predicted (*Ma et al., 2016b*). Evidence was found that globally detected calcium signals are linked to hemodynamic functional connectivity and that the spatial information of the cortical network functional connectivity may be embedded in the phase of global calcium waves (*Matsui et al., 2016*). Nevertheless, it has not been investigated until now, which cortical network activity reflected by the hemodynamic BOLD signal underlies the active phase of the slow wave rhythm – the slow wave event.

In order to investigate brain-wide BOLD correlates of locally occurring slow waves, these two signals have to be recorded simultaneously, and slow waves have to be detected in a spatiotemporally precise manner and directly related to BOLD activity.

To achieve this aim, optic fiber-based calcium recordings are well-suited as they can be performed unperturbed by the magnetic field of the MR scanner (*Schmid et al., 2016*; *Schulz et al., 2012*).

In this study, we examine BOLD hemodynamic responses upon individually detected, optically recorded slow calcium waves by employing event-related fMRI analysis which can be used to identify BOLD changes in response to individual events. The key concepts of event-related analyses are time-locking and signal averaging. Although originally intended to reveal transient changes in brain activation associated with the presentation of discrete sensory stimuli, event-related analysis can detect BOLD activity upon any type of previously defined events (*Huettel et al., 2009*; *Josephs et al., 1997*), in our case the active phases of the optically recorded slow calcium wave activity. In addition to the general linear model (GLM) based event-related method, we employ two multivariate exploratory approaches for fMRI data: seed-based and independent component analysis (ICA).

Here we show that a change of excitability state of a small population of neurons can be indicative for rather global network states being reflected by whole-brain BOLD activity patterns, demonstrating the particular relevance of considering readouts of local population activity for fMRI measurements. We reveal the interrelation of a neurophysiological defined slow wave event and a macroscopic, network organizing signal, and find a cortex-wide BOLD fMRI correlate.

## Results

### Optic-fiber based recordings are suitable for recording slow oscillation-associated calcium waves

We employed an optic fiber-based approach to detect calcium transients simultaneously to fMRI scans within a 9.4 T small animal MR scanner (*Figure 1A*) (*Schmid et al., 2016*; *Schulz et al., 2012*). To monitor intracellular calcium as proxy of spiking, the synthetic dye Oregon Green 488 BAPTA-1 (OGB-1) was injected stereotactically resulting in a column-like stained region with a diameter of about 600 µm using the multicell bolus loading technique (*Stosiek et al., 2003*) (*Figure 1B,C*). The tip of the optical fiber with a diameter of 200 µm was implanted dorsal to the stained region (*Figure 1B*) in primary somatosensory cortex (S1 front limb, S1FL). As mentioned above, OGB-1 stains both neurons and astrocytes, as confirmed by co-staining with the astrocytic marker SR101 (*Nimmerjahn et al., 2004*) (*Figure 1D*). In addition, we employed the genetically encoded calcium

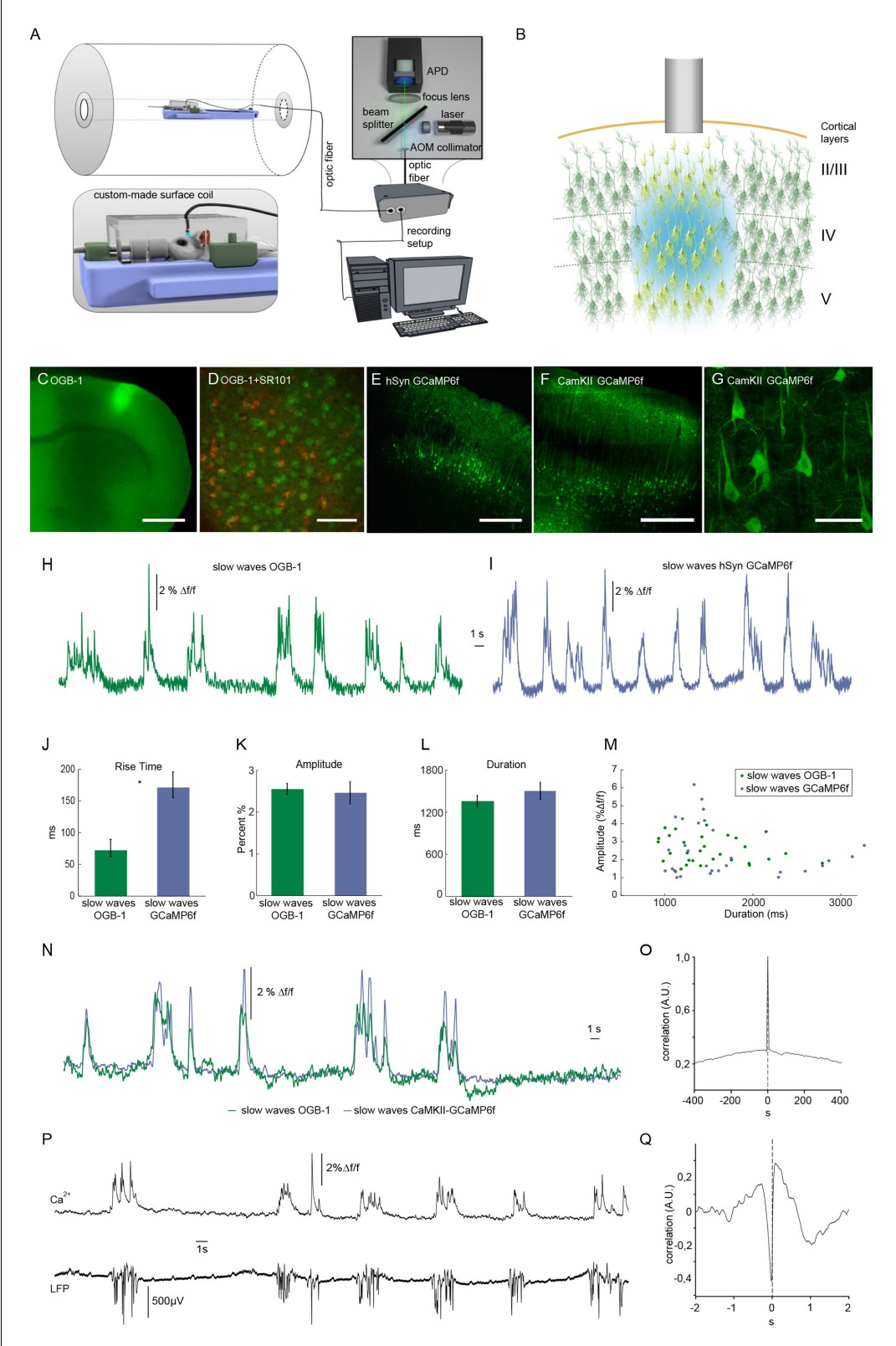

**Figure 1.** Optical recordings of slow oscillation-associated calcium waves within a small animal MR scanner in anesthetized rats. (A) Scheme of optical recording setup comprising of a solid state laser (488 nm) coupled to a multimode optic fiber with a diameter of 200 µm. Emitted light is recorded by an avalanche photo diode (APD) and converted to a voltage signal. For simultaneous fiber-based calcium recordings and fMRI, a custom-built surface coil with a lead-through for the optic fiber is employed. (B) Upon bolus loading with OregonGreen BAPTA-1 (OGB-1) or after viral transduction with

*Figure 1 continued on next page*

*Figure 1 continued*

GCaMP6f, an optic fiber (grey shaded rectangle) is implanted in primary somatosensory cortex to record an integrated, yet spatially defined neural population signal. Cortical layers are indicated. (C) Fluorescence micrograph of a coronal brain slice depicting the area of OGB-1 staining at the level of primary somatosensory cortex, scale bar = 1 mm. (D) Fluorescence micrograph of co-staining with OGB-1 (green) and SR101 (red) (scale bar = 100 µm) (E) Expression of GCaMP6f four weeks after injection of AAV1.Syn.GCaMP6f (scale bar = 500 µm). (F–G) Expression of GCaMP6f four weeks after injection of AAV1.CamKII.GCaMP6f scale bar F = 500 µm G = 50 µm. (H) Calcium trace upon OGB-1 staining recorded in the MR scanner during slow wave activity. Optically recorded spontaneous stereotypical calcium waves are interrupted by periods of network quiescence. (I) Optical recordings in GCaMP6f expressing rats similarly reveal slow calcium waves. (J–M) Quantifications of slow calcium wave parameters in OGB-1 stained vs. GCaMP6f expressing rats reveal significant differences only in rise times (p=0.0001), whereas amplitude and durations are not significantly different (p=0.13; p=0.61). (N) Simultaneous optical calcium recordings outside the MR scanner using two optic fibers implanted into left hemisphere with OGB-1 staining (green trace) and with CaMKII-GCaMP6f staining. An overlay of recorded traces shows that with both indicators the occurrence of a slow wave is reflected by similar transients. (O) Cross-correlation of simultaneously recorded optical calcium signals obtained in (N) indicates high similarity between both signals. (P) Simultaneous optical calcium and LFP recordings outside of the MR scanner indicating correlation of slow calcium waves with electrically recorded slow oscillations under isoflurane anesthesia. (Q) Cross-correlation of optically and electrically recorded traces result in a prominent peak with almost zero time shift, indicating a close similarity of the two signals.

DOI: https://doi.org/10.7554/eLife.27602.003

The following source data and figure supplement are available for figure 1:

**Source data 1.** Amplitude, duration and rise time values for quantification of OGB-1 slow calcium wave events (data shown in *Figure 1J–M*).
DOI: https://doi.org/10.7554/eLife.27602.005
**Source data 2.** Amplitude, duration and rise time values for quantification of GCaMP6f slow calcium wave events (data shown in *Figure 1J–M*).
DOI: https://doi.org/10.7554/eLife.27602.006
**Figure supplement 1.** Immunofluorescent characterization of GCaMP6f expressing neurons.
DOI: https://doi.org/10.7554/eLife.27602.004

indicator GCaMP6f under control of a pan-neuronal promoter hSyn (*Figure 1E*) and the CamKII promoter limiting expression to excitatory neurons (*Figure 1F,G*, *Figure 1—figure supplement 1*).

Experiments were conducted under isoflurane anesthesia (1.1–1.8%) inducing and maintaining slow wave activity with recurrent slow wave events at similar frequencies (*Chen et al., 2013b*; *Grienberger et al., 2012*; *Lissek et al., 2016*; *Luczak et al., 2007*; *Stroh et al., 2013*). Calcium waves were reliably recorded during simultaneous fMRI recordings (8 out of 10 experiments in 7 out of 8 animals), occurring at frequencies ranging between 9 and 15 events/min (mean 10.9 ± 1.3 events/min), in line with previous in vivo recordings (*Busche et al., 2015*; *Kerr et al., 2005*; *Stroh et al., 2013*) (*Figure 1H,I*).

For OGB-1, the typical and rather uniform calcium waves exhibited sharp rise times (72 ± 14 ms, and rather variable durations and amplitudes of 1356 ± 85 ms and 2.6 ± 0.1 Δf/f, respectively (n = 30 traces, five animals). For GCaMP6f (hSyn) the rise times exhibited slower kinetics (163 ± 14 ms, n = 30 traces, two animals, *Figure 1J*) presumably due to four calcium binding sites of the calmodulin, which need to be occupied to reach maximum fluorescence change, while similar values for durations and amplitudes of slow waves were observed as compared to OGB-1 (*Figure 1K,L,M*). To test, whether indeed OGB-1 and GCaMP6f both reveal the same slow wave events, we conducted two-fiber experiments outside of the MR scanner, in two animals expressing GCaMP6f under control of the CaMKII promoter in S1FL, and being injected with OGB-1 2 mm posterior to that site on the same hemisphere. (*Figure 1N*). Indeed, we found the associated slow waves to be highly correlated (*Figure 1N,O*). The calcium waves were separated by periods of network quiescence of variable durations (*Figure 1H,I,N,P*), equivalent to down states in electrophysiological recordings (*Poulet and Petersen, 2008*; *Seamari et al., 2007*; *Steriade et al., 1993c*; *1993b*). In line with previous results (*Busche et al., 2015*; *Stroh et al., 2013*), the observed population calcium transients were correlated with electrically recorded slow oscillations in the local field potentials (LFPs; *Figure 1P,Q*).

## Onsets and durations of optically recorded slow calcium waves can be utilized for hemodynamic response extraction and event-related fMRI analysis

Calcium slow wave events were identified using an adapted version of an algorithm established for the detection of slow oscillations in electrophysiological recordings in vitro and in vivo (*Seamari et al., 2007*), separating slow oscillatory activity and silent periods based on exponential

moving average (EMA) filters. We adapted and optimized this procedure for the identification of slow wave activity in optical calcium recordings in vivo. Applying this detection algorithm yielded an average number of 298 ± 20 slow wave events per 30 min experiment (n = 7 animals; *Figure 2B*). The resulting onsets and durations of calcium slow waves were encoded as a binary array of events (one) and silent periods (zero). These 'slow wave vectors' (*Figure 2A,C*) were used as a condition in an event-related fMRI analysis and were convolved and correlated with the fMRI signal to detect BOLD responses following the specified events defined in a design matrix.

The temporal characteristics of the hemodynamic response (HR) elicited by slow wave activity remains insufficiently explored and due to the specific slow wave network activity may differ from HRs elicited e.g. by sensory stimulation. Therefore we first examined the shape of typical HRs elicited by individual slow wave events in cortical ROIs on left and right hemisphere (*Figure 2D*, *Figure 2—figure supplement 1*). Timecourse of averaged HRs exhibited a sharp rise from baseline to maximum during 0.7 s ± 0.4 s after the onset of a slow calcium wave, reaching the maximum signal amplitude of 0.17%±0.03% (ΔSA) at 6.5 s ± 0.4 s (TTP) with a half maximum duration (HMD) of 6.8 s +- 0.5 s. Next, we extracted HRs following the onsets of slow calcium waves from the 30 most active voxel of each measurement using a model-free approach based on a finite impulse response method (FIR) (*Dale, 1999*; *Glover, 1999*) which approximates the underlying timecourse of HRs without any *a priori* assumptions (*Jansma et al., 2013*) (*Figure 2E*).

Subsequently, we used a leave-n-out approach analyzing datasets with averaged HRs extracted solely from the remaining animals, to avoid circular analysis (*Kriegeskorte et al., 2009*) (*Figure 2F*). For this purpose, the resulting HRs were averaged and used as hemodynamic response functions (HRFs) convolved with the event arrays generated from the calcium waves to create activation maps reflecting ongoing BOLD activity during slow oscillation-associated slow wave activity (*Figure 2G*).

## Calcium slow wave event-related analysis reveals cortex-wide BOLD correlate

We used event-related analysis (*Josephs et al., 1997*), based on either the extracted HRF or the FIR-derived HRF (see Materials and Methods) to correlate onsets and durations of locally detected slow calcium waves (*Figure 3A,B*) with the simultaneously acquired fMRI BOLD signal. Event-related fMRI analysis accounts for responses to each predefined event which is specified in terms of its onset and duration (*Huettel et al., 2009*).

Using the detected individual slow wave events in the calcium signal in an event-related analysis based on extracted HRFs revealed BOLD activation of nearly the entire cortex during slow wave activity (*Figure 3C,D*; *Figure 3—figure supplement 1*) in 6 of 8 experiments (n = 7 rats). Although cluster sizes varied over animals and experiments, the activation clusters were spanning almost all slices for all experiments showing BOLD activation (*Figure 3C,D*; *Figure 3—figure supplement 2*). While activation patterns were occasionally patchy (*Figure 3—figure supplement 1C*; *Figure 3—figure supplement 2*), each cortical area was showing at least partial BOLD activation predicted by slow calcium waves in every experiment, i.e. no cortical area was excluded from BOLD activity.

We next asked, whether the BOLD activity quantitatively differs between cortical areas. Therefore, we analyzed bilateral ROIs drawn on the F-maps derived from the event-related analysis employing HRFs obtained from a FIR analysis previously performed in a different set of animals, both in somatosensory and visual cortex (*Figure 3E*). The F-values of the two regions of the cortex showed no significant differences in mean values (*Figure 3F*). In addition, we extracted the HR of the two regions from the beta-maps, no apparent differences in the shape of the HRs could be observed (p=0.23, Wilcoxon rank-sum test), indicative of a rather uniform activation profile throughout the cortex. These results suggest that locally recorded slow calcium wave events have a cortex-wide correlate in the fMRI BOLD signal representing global slow wave activity.

To confirm that the resulting HRs do not differ depending on the dataset used for extraction, we performed a cross-validation procedure and analyzed datasets with two different previously extracted HRs and compared cluster sizes and T-values (*Figure 2—figure supplement 3*). For this analysis we used mean HRs extracted from at least two other animals.

Next, we compared our individualized HRF estimation based on the experimental slow wave data with the canonical HRF, routinely employed in fMRI data both in humans (*Glover, 1999*; *Lindquist and Wager, 2007*) and rats (*Amirmohseni et al., 2016*), by applying the SPM standard procedure which uses a continuous canonical HRF model (*Figure 3—figure supplement 2A,B*). We

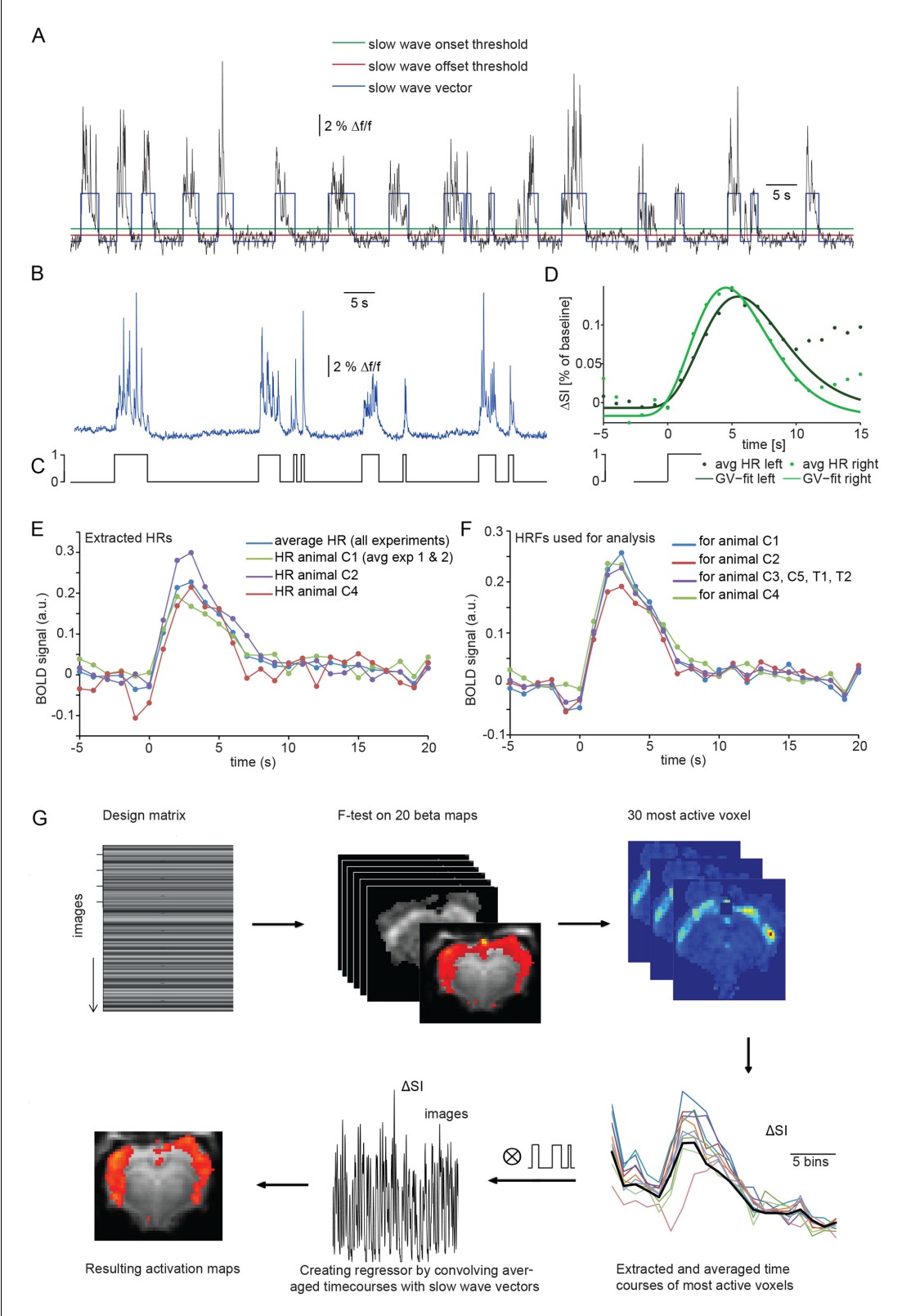

**Figure 2.** Identifying a BOLD response upon slow wave events detected in cortical calcium recordings. (A) Onset and termination thresholds based on EMA filters detect slow wave events in a calcium trace recorded in the MR scanner and can be used to create a slow wave vector. (B,C) The binarized slow wave vector (C) containing the precise timing of the slow calcium wave events (B) can be used as an event array for subsequent fMRI analysis. (D) Averaged BOLD signal in cortical ROIs in both hemispheres of an individual experiment upon the onset of slow calcium wave events, fitted with a

*Figure 2 continued on next page*

*Figure 2 continued*

gamma variate (GV). Note, that due to subsequent slow calcium wave events within the time frame of the respective BOLD response (see A,B), the raw signal does not decay to baseline levels. (E) Extracted hemodynamic responses (HRs) with finite impulse response (FIR) methods. Intra- and interindividual timecourses show almost identical signal characteristics. (F) Individual HRs were averaged and used as HRF for subsequent fMRI analysis (for animal C1 HRs from animal C2 and animal C3 were averaged; for animal C2 HRs from animal C1 and animal C3; for animal C3 HRs from animal C1 and C2; for animal C4 HRs from animal C1, C2; for animal C3, C5, T1 and T2 HRs from animal C1, C2 and C4 were averaged). (G) Flow chart of fMRI analysis procedure starting with the design matrix, defined by the FIR basis function set and the spontaneous calcium wave as event.

DOI: https://doi.org/10.7554/eLife.27602.007

The following figure supplements are available for figure 2:

**Figure supplement 1.** Comparison of HR extracted in the 30 most active voxel and in ROI placed in a cortical area.
DOI: https://doi.org/10.7554/eLife.27602.008
**Figure supplement 2.** Number of active voxel depends on the threshold for calcium wave detection.
DOI: https://doi.org/10.7554/eLife.27602.009
**Figure supplement 3.** Cross-validation procedure to test for impact of respective dataset used for HRF extraction.
DOI: https://doi.org/10.7554/eLife.27602.010

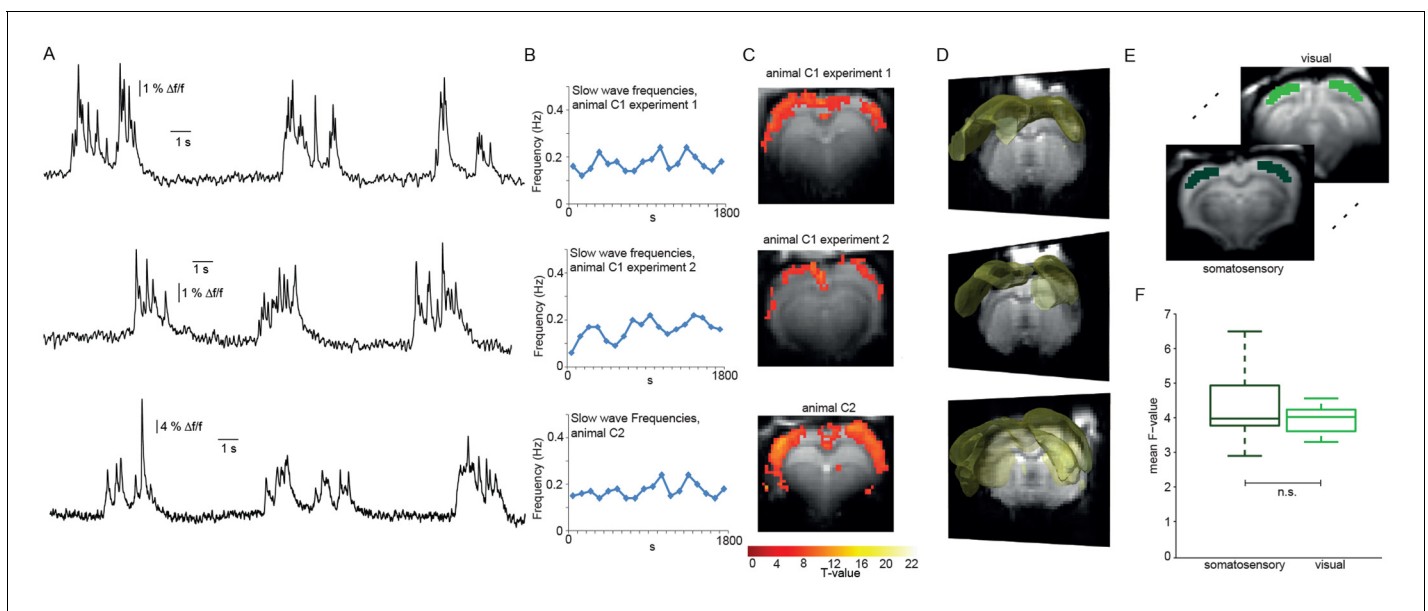

**Figure 3.** Relating neurophysiologically defined slow calcium wave events to fMRI BOLD signal reveals cortex-wide activation. (A) Excerpts of calcium traces acquired during fMRI measurements in three experiments depicting slow calcium waves. (B) Frequencies of slow calcium waves within an imaging experiment. (C) BOLD correlate upon FIR-based event-related fMRI analysis in the cortex. (D) 3D renderings depict cortex-wide expansion of activity shown in (B) (light-yellow shading shows above threshold activity). (E,F) Quantitative comparison of BOLD activation in somatosensory (10 ROIs, 4 animals, 5 experiments) and visual cortex (6 ROIs, 4 animals, 5 experiments). No significant difference in the mean F-values could be detected (mean F-values in somatosensory cortex: $4.3 \pm 0.3$, in visual cortex: $4.0 \pm 0.1$, p=0.41).
DOI: https://doi.org/10.7554/eLife.27602.011

The following source data and figure supplements are available for figure 3:

**Source data 1.** Cluster sizes and T-values for experimental HRF versus canonical HRF for all experiments (Table for data shown in *Figure 3—figure supplement 2*).
DOI: https://doi.org/10.7554/eLife.27602.015
**Figure supplement 1.** BOLD correlate upon FIR-based event-related fMRI analysis of additional datasets.
DOI: https://doi.org/10.7554/eLife.27602.012
**Figure supplement 2.** Similar activity patterns upon employing HRFs extracted from datasets based on FIR approach and classical canonical HRF model.
DOI: https://doi.org/10.7554/eLife.27602.013
**Figure supplement 3.** Slow wave BOLD correlates depend on precise slow calcium wave dynamics detected in individual experiments.
DOI: https://doi.org/10.7554/eLife.27602.014

compared T-values in the previously defined ROIs and the approaches yielded comparable results (*Figure 3—source data 1*).

To further test for the specificity and interrelation of a recorded calcium event array to the respective BOLD activation and to rule out a general effect of specifically spaced event-timing causing the reported effects, we applied two control procedures for this analysis: swapping regressors between the animals (*Figure 3—figure supplement 3A,B*) and temporally reverting the event arrays (*Figure 3—figure supplement 3C,D*) resulted in no BOLD activation in any of the data sets, thus confirming the specificity of the observed activations.

## Slow oscillation-associated calcium waves in the thalamus

As mentioned, slow oscillations-associated calcium waves recruit not only the cortex, but also the thalamus. We therefore conducted optic fiber-based calcium recordings in the posterior medial nucleus of the somatosensory thalamus (POm; *Figure 4A*). As expected, we were able to record typical slow wave events, albeit at a lower SNR (*Figure 4B*) in two animals, with similar characteristics compared to the cortically recorded slow waves, in line with a previous study (*Stroh et al., 2013*). We then extracted the HR as described earlier. Using those thalamic calcium waves as events, we found BOLD activation in the cortex only (*Figure 4E,F*; 2 experiments in two animals), similar to the cortical recordings. This confirmed the tight synchronicity between cortical and thalamic calcium

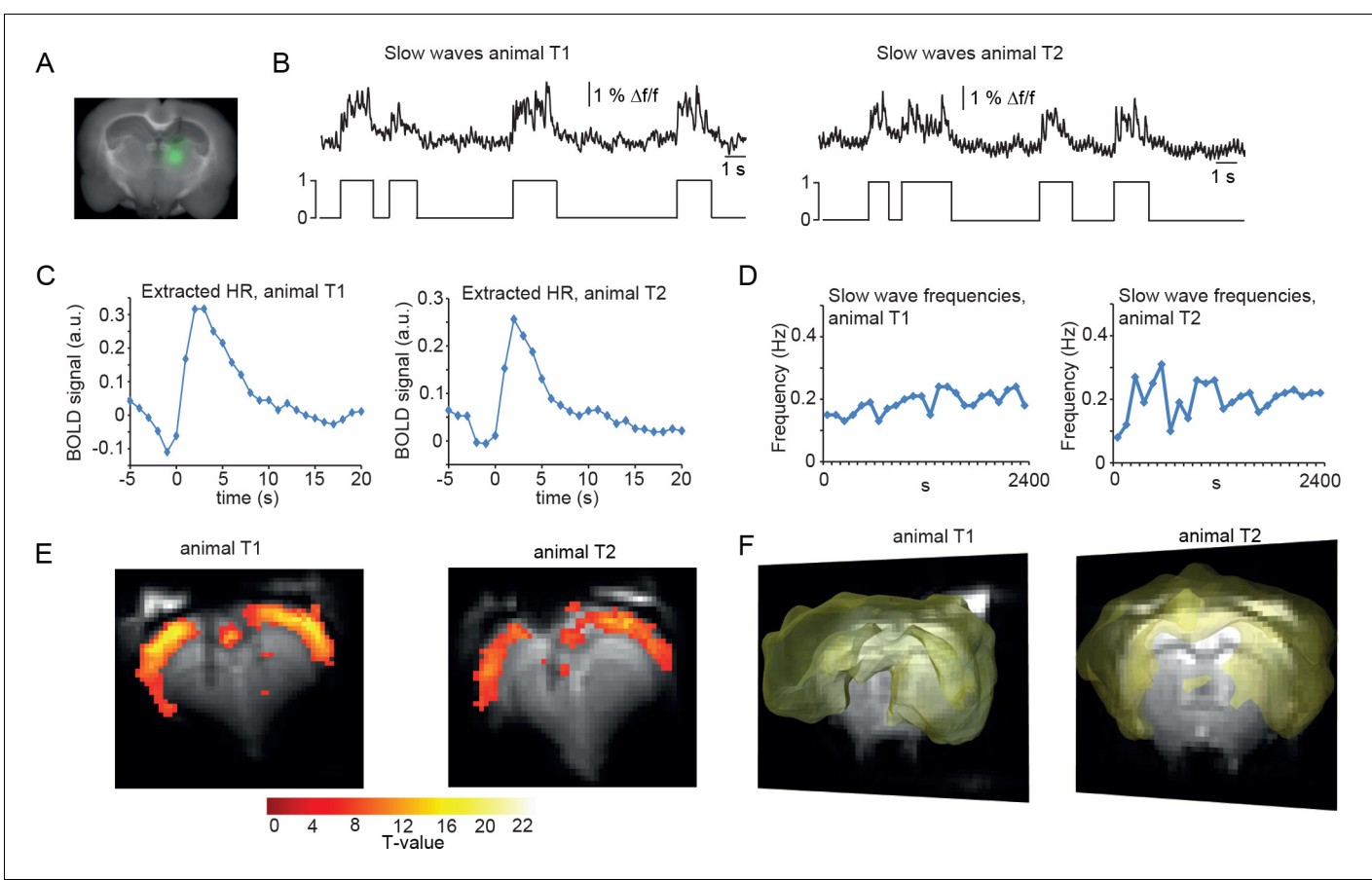

**Figure 4.** BOLD correlate upon FIR-based event-related fMRI analysis using slow wave vectors acquired in the thalamus. (A) Micrograph of a coronal brain slice depicting the area of OGB-1 staining at the level of thalamus (POm). (B) Excerpt of calcium trace acquired in thalamus during fMRI measurements depicting slow calcium waves and the binarized slow wave vector derived from these calcium waves. (C) Extracted hemodynamic responses (HRs) with FIR methods. The time course shows almost identical signal characteristics as in previous datasets using cortical calcium recordings. (D) Frequencies of slow calcium waves within an imaging experiment. (E) BOLD correlate upon FIR-based event-related fMRI analysis in the cortex of animal T1 and T2. (F) 3D rendering depicts cortex-wide expansion of activity shown in (E) (light-yellow shading shows above threshold activity).
DOI: https://doi.org/10.7554/eLife.27602.016

waves. The fact that no BOLD signal in thalamus was detected is most likely due to using a MR surface coil which results in lower detection sensitivity in subcortical regions.

## Cortex-wide BOLD activation is specific for slow calcium wave events

GLM-based fMRI analysis identifies voxel correlated with the model timecourse. Model-free analysis approaches do not require a priori knowledge about activation patterns and do not consider voxel individually to reveal potential voxel interactions. Consequently, in addition to the event-related GLM method, we employed independent component analysis (ICA) for ongoing slow wave activity. ICA aims to recover a set of maximally independent sources from their observed mixtures without knowledge of the source signals or the mixing parameters (*Hyvärinen and Oja, 2000*). With this analysis, during ongoing slow wave activity, we found ICA components mirroring the results obtained by using the slow calcium waves as regressor in the same animals (*Figure 5A*). Notably, the timecourse of the ICA component revealing cortex-wide activation is highly correlated with the GLM regressor obtained from taking into account slow wave onsets detected in the calcium recordings (R = 0.4 ± 0.05, n = 9 animals; *Figure 5B*), further confirming the neurophysiological basis of this component. To test for specificity of those results we also calculated the same cross-correlation of the ICA component timecourse with the GLM-regressor, but with the GLM-regressor obtained from the temporally mirrored slow wave vector (see *Figure 3—figure supplement 3C*) of the same animal, which leads to a complete absence of correlation between the ICA component timecourse and the event-related GLM-regressor (*Figure 5C*).

As ICA decomposition can produce a substantial number of noise components, we extracted the component set (number of components identified by minimum description length criterion (MDL), see Materials and Methods) for a group ICA (*Figure 5—figure supplement 1A*), as well as for the corresponding single datasets (*Figure 5—figure supplement 1B*). Notably, we could again identify pan-cortical activation patterns in both the group- and the single-subject based results (ICs #3, #6 in *Figure 5—figure supplement 1*). In addition, with this analysis we could identify components related to typical resting state networks including auditory (IC #1) and visual (IC #8) cortex, striatum (IC #7) and hippocampus (IC #20) in line with others (*Jonckers et al., 2011*). A comparison of the power spectra related to the timecourses of the pan-cortical and two other components shows that the pan-cortical component exhibits higher power in the <0.1 Hz range (*Figure 5—figure supplement 2A–D*).

Regressing out the timecourse derived by the slow wave vector obtained from the calcium transients (see Materials and Methods), leads to complete absence of pan-cortical components in the ICA (*Figure 5—figure supplement 2E,F*).

Next, we asked, whether the cortex-wide BOLD activation is indeed specifically related to individually detected slow wave events. Therefore, we contrasted this activity, maintained by isoflurane anesthesia, with persistent activity, maintained by medetomidine sedation. This persistent activity is lacking slow calcium wave activity, i.e. bimodality (*Figure 5—figure supplement 3*). We confirmed the differential population activity under the two conditions by assessing both spontaneous as well as sensory-evoked calcium transients as previously described separately for both conditions (*Schmid et al., 2016*; *Schulz et al., 2012*; *Stroh et al., 2013*).

During persistent activity, we did not find any component indicative of a synchronous cortical activation (n = 6), but we found resting state networks as previously described (*Figure 5—figure supplement 4A,B*) (*Hsu et al., 2016*; *Jonckers et al., 2011*; *Kalthoff et al., 2013*; *Lu et al., 2012*; *Ma et al., 2016b*).

To complement the ICA approach, we employed seed-based analyses. Using the somatosensory cortex as seed-ROI, during ongoing slow wave activity, we found a correlating cortex-wide BOLD activation (*Figure 6A*). Furthermore, the average signal in the somatosensory seed-ROI was strongly correlated with the normalized signal of the pan-cortical component previously revealed by the ICA (*Figure 6B*; *Figure 6—source data 1*). During persistent activity this pan-cortical activity was absent (*Figure 6—figure supplement 1*; *Figure 6—source data 1*), but bilateral somatosensory activity, potentially reflecting inter-hemispheric connectivity between the somatosensory cortices (*Hodkinson et al., 2016*) could be detected (*Figure 6—figure supplement 1*).

As we occasionally observed hippocampal BOLD (*Figure 3—figure supplement 1C*; *Figure 3—figure supplement 2A,B*) we applied the same seed-based approach to the hippocampal formation. Again, only during ongoing slow wave activity we found pan-cortical BOLD activation related to the

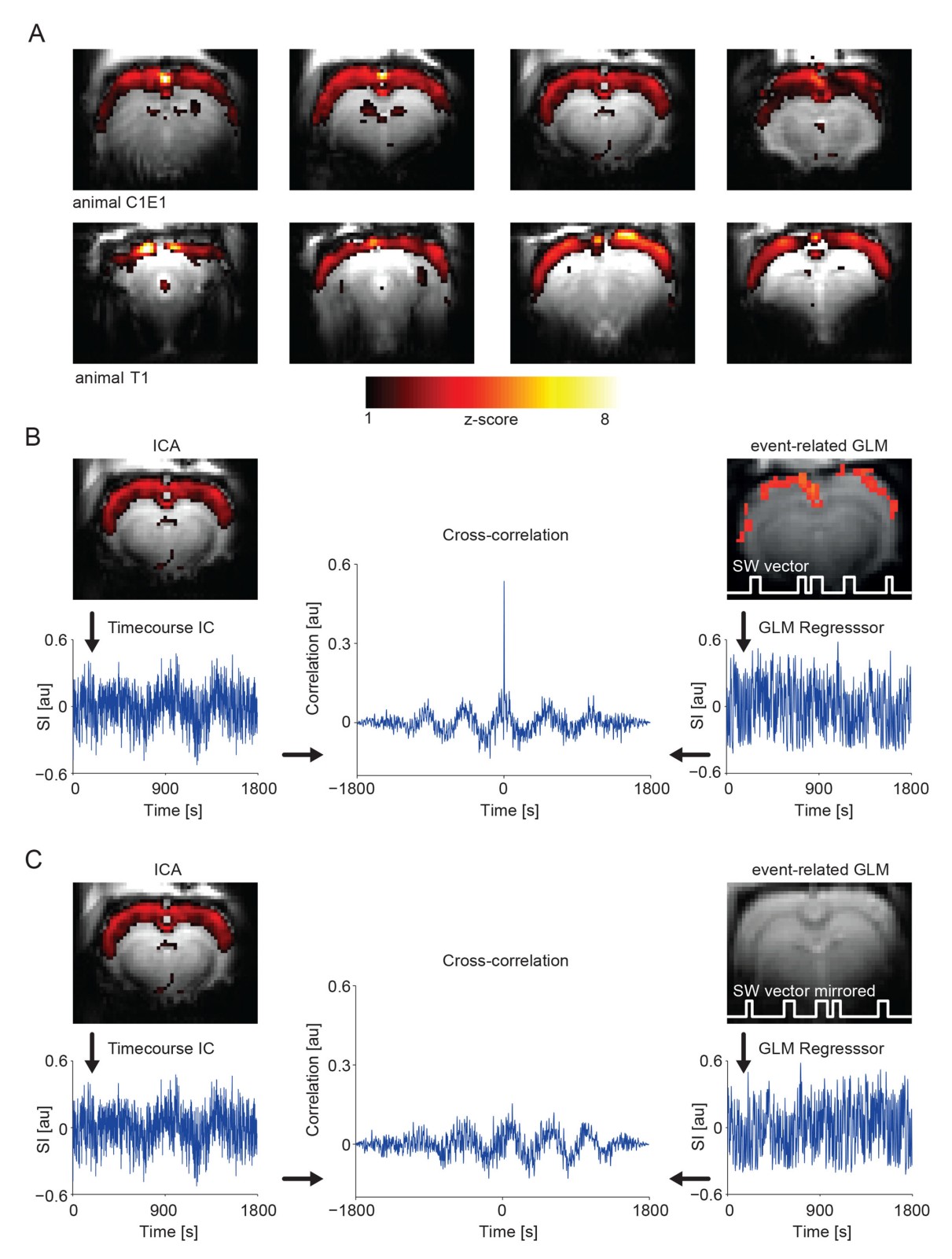

**Figure 5.** IC analysis in slow activity results in a component showing pan-cortical activation specifically correlating to the calcium slow wave vector. (**A**) Pan-cortical component in slow wave activity (isoflurane anesthesia). Spatial correlate of two exemplary ICs showing consistent high values covering the entire cortex. Two representative animals and four imaging slices each are depicted. (**B**) The timecourse of the pan-cortical activation component is highly correlated with the regressor extracted from the slow wave vector (SW vector) obtained from the calcium recordings (r = 0.535 and r = 0.628) for

*Figure 5 continued on next page*

*Figure 5 continued*

upper and lower ICs shown in (**A**), when this SW vector is mirrored in time, this correlation completely disappears (**C**). (**B**) and (**C**) both shown for animal C1E1.

DOI: https://doi.org/10.7554/eLife.27602.017

The following figure supplements are available for figure 5:

**Figure supplement 1.** Complete component set of IC analysis in slow wave activity.

DOI: https://doi.org/10.7554/eLife.27602.018

**Figure supplement 2.** Pan-cortical BOLD appears only in ICs with higher power in lower frequencies of their time courses' power spectra.

DOI: https://doi.org/10.7554/eLife.27602.019

**Figure supplement 3.** Neuronal response patterns upon sensory stimulation for slow wave and persistent activity.

DOI: https://doi.org/10.7554/eLife.27602.020

**Figure supplement 4.** IC Analysis showing typical resting state networks during persistent activity.

DOI: https://doi.org/10.7554/eLife.27602.021

hippocampal seed-ROI and also found a correlation to the pan-cortical ICA component (*Figure 6—figure supplement 2A–C*, *Figure 6—source data 2*).

Finally, we conducted an optical calcium recordings-informed correlation analysis (*Figure 6C*). When HRs upon slow calcium wave onsets in a ROI in somatosensory cortex are correlated with every remaining voxel in the brain, again a cortex-wide network becomes apparent. Independent of the location of the somatosensory ROI (right or left hemisphere) this correlation analysis revealed highest r-values for the entire cortex. Additionally, correlation values reveal that slow wave related BOLD activity recruits almost all cortical areas with nearly synchronous time-to-peak values for HRs (*Figure 6D*; *Figure 6—figure supplement 3*).

## Discussion

In summary, our results demonstrate that locally recorded calcium events related to slow wave activity show a global cortical fMRI BOLD correlate. The novelty of our approach consists in establishing a direct relation of neurophysiologically defined slow wave events to the macroscopic fMRI signal by detecting the precise time points of individual slow waves and probing their underlying hemodynamic responses by using event-related fMRI analysis.

### Slow wave activity in the context of large-scale networks

Our results provide evidence for large-scale recruitment of cortical networks upon a specific type of neural activity – slow waves – which have previously been shown to be present under different types of anesthesia (*Busche et al., 2015*; *Chauvette et al., 2011*; *Petersen et al., 2003*; *Sanchez-Vives et al., 2017*; *Sanchez-Vives and McCormick, 2000*; *Seamari et al., 2007*; *Stroh et al., 2013*; *Zucca et al., 2017*), and in natural slow wave sleep (*Massimini et al., 2004*; *Sanchez-Vives et al., 2017*; *Steriade et al., 1993a*; *Steriade et al., 1993b*; *Steriade et al., 1993c*; *1993b*; *Steriade and Timofeev, 2003*; *Steriade et al., 2001*). Although the main features appear similar to those of the SWS oscillation (*Destexhe et al., 2007*; *Sanchez-Vives et al., 2017*), under anesthesia, slow waves appear more rhythmic and more synchronous across the cortex and also show longer silence periods (*Busche et al., 2015*; *Chauvette et al., 2011*).

During natural sleep, spindle activity - which is related to memory consolidation - is grouped by slow waves (*Mölle et al., 2002*), but this is not the case for anesthesia induced slow waves (*Murphy et al., 2011*). Nevertheless sleep slow waves and anesthesia slow waves may recruit the same cortical and subcortical structures (*Murphy et al., 2011*). Although cortically generated (*Steriade et al., 1993c*; *Timofeev et al., 2000*) slow waves also engage the thalamus (*Sheroziya and Timofeev, 2014*; *Steriade et al., 1993a*; *Stroh et al., 2013*) and hippocampus (*Busche et al., 2015*; *Ji and Wilson, 2007*; *Sirota et al., 2003*), suggesting that slow wave associated excitation plays a synchronizing role in functional coupling of remote brain regions (*Hahn et al., 2006*). Indeed we occasionally observed hippocampal BOLD activation during slow waves and therefore conducted a seed-based analysis for an atlas-based hippocampal ROI, showing that voxel in the hippocampus correlate well with cortical voxel, again spanning the entire cortical surface during ongoing slow wave activity. This correlation of hippocampal activity to slow-wave

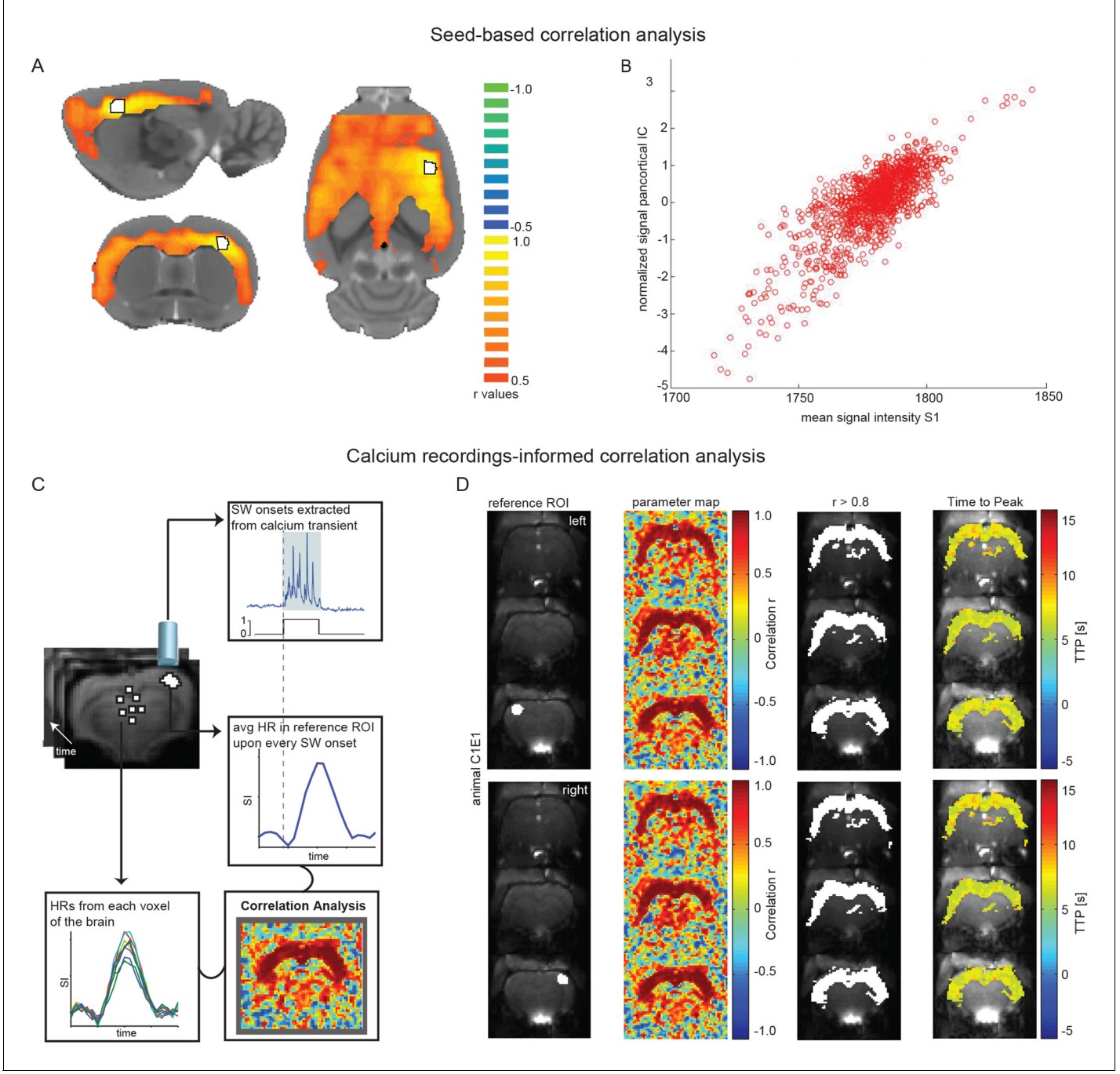

**Figure 6.** Seed-based and calcium recordings-informed correlation analyses reveal pan-cortical BOLD activation during slow wave activity. (**A**) Signal correlation between primary somatosensory cortex ROI (S1; white with black outline) and the rest of the cortex in a representative animal during slow wave activity. (**B**) Correlation between the average signal in S1 and the pan-cortical independent component (IC) from this representative animal's ICA (pearson correlation coefficient, r = 0.799, p<0.001). (**C**) Flow-chart of correlation analysis procedure employing slow wave onsets from calcium transients. (**D**) ROIs on both hemispheres (ipsi- and contralateral sites of the fiber recordings), parameter maps showing r-values and regions of highest correlation (r > 0.8; white). The mean HR was calculated from these thresholded regions and time to peak of voxel-by-voxel HR was determined.
DOI: https://doi.org/10.7554/eLife.27602.022

The following source data and figure supplements are available for figure 6:

**Source data 1.** Values for seed-based correlation between S1 and cortex during slow wave and persistent activity (n = 4 animals; table for data shown in *Figure 6* and *Figure 6—figure supplement 1*).
DOI: https://doi.org/10.7554/eLife.27602.026
*Figure 6 continued on next page*

*Figure 6 continued*

**Source data 2.** Values for seed-based correlation between hippocampal formation (hf) and cortex during slow wave and persistent activity (n = 4 animals; table for data shown in *Figure 6—figure supplement 2*).
DOI: https://doi.org/10.7554/eLife.27602.027
**Figure supplement 1.** Seed-based correlation analysis during persistent activity.
DOI: https://doi.org/10.7554/eLife.27602.023
**Figure supplement 2.** Seed-based correlation analysis originating from hippocampal formation (hf).
DOI: https://doi.org/10.7554/eLife.27602.024
**Figure supplement 3.** Calcium recordings-informed correlation analysis reveals pan-cortical BOLD activation during slow wave activity.
DOI: https://doi.org/10.7554/eLife.27602.025

related cortical BOLD (*Chan et al., 2017*) might provide a mechanism of coordination for reactivation and redistribution of hippocampus-dependent memories to neocortical sites (*Buzsáki, 1996*; *Mitra et al., 2016*). Although this is unlikely the case for anesthesia related slow waves (*Murphy et al., 2011*), they might recruit the same structures and thereby provide a spatiotemporal mechanism for such plasticity-relevant cortico-hippocampal functions (*Logothetis et al., 2012*).

Although the thalamus is being recruited during slow wave activity (*Sheroziya and Timofeev, 2014*; *Stroh et al., 2013*), we did not observe robust thalamic BOLD activity, despite of optically recorded thalamic slow calcium waves. This might be due to a lower density of neurons carrying the slow wave rhythm, reflecting the lower overall cell density in the thalamus in comparison to the cortex (*Meyer et al., 2013*) or due to reduced sensitivity of the MR surface coil in deeper brain areas.

Intracortically, slow waves can propagate as a travelling wave (*Busche et al., 2015*; *Matsui et al., 2016*; *Stroh et al., 2013*), likely by relays of local populations of cells (*Destexhe et al., 2007*). Depending on excitability state of the network, a slow wave event could either remain local (*Nir et al., 2011*; *Vyazovskiy et al., 2011*) or could spread 'smoothly like an oil-spot' (*Massimini et al., 2004*; *Sanchez-Vives et al., 2017*). The recruitment of sub-cortical regions is discussed to rather involve long-range excitatory projections (*Leong et al., 2016*).

Although slow wave activity is a multiscale phenomenon (*Jercog et al., 2017*; *Sanchez-Vives et al., 2017*), previous work has focused either on local and cellular (*Chen et al., 2013b*; *Grienberger et al., 2012*; *Kerr et al., 2005*; *Sanchez-Vives and McCormick, 2000*; *Stroh et al., 2013*) or on large-scale characteristics (*Massimini et al., 2004*) of slow waves. Here, we bridge these mentioned levels of observation by relating the mesoscopic representation of slow waves in a defined cortical or thalamic population - in a spatio-temporally precise manner - to the whole-brain hemodynamic BOLD signal.

## Specific relation of the cortex-wide BOLD signature to slow wave activity

We contrast our findings obtained during slow wave activity to a network activity, maintained by light sedation, in which slow calcium wave events are absent. Indeed, the calcium recordings reveal distinct response properties of local sensory networks, matching earlier studies under the same condition (*Schmid et al., 2016*; *Schulz et al., 2012*).

Here, seed-based analysis and ICA revealed typical default mode and resting state networks as described previously (*Hsu et al., 2016*; *Lu et al., 2012*). Notably, we did not find any component resembling pan-cortical activation as identified for slow wave activity, speaking for a direct relation of slow waves to this BOLD pattern.

Furthermore, during slow wave activity, we could demonstrate that the ICA component showing pan-cortical activation is highly correlated to the GLM timecourse derived from optically detected calcium waves. Exclusively this pan-cortical component is related to higher power in the ultra-low frequency range. Such low frequency components were related to slow wave activity in spontaneous EEG/fMRI signal fluctuations in earlier studies and have been investigated in different species (*Leopold et al., 2003*; *Schölvinck et al., 2010*; *Steriade and Amzica, 1998*; *Wang et al., 2012*; *Wen and Liu, 2016*)) including humans (*Achermann and Borbély, 1997*; *Boly et al., 2012*; *Dang-Vu et al., 2008*; *Hiltunen et al., 2014*; *Marshall et al., 2006*; *Massimini et al., 2004*; *Wen and Liu, 2016*). Notably, if the timecourse derived from the slow wave vector is regressed out of the data, the ICA is devoid of any components showing pan-cortical activation.

Nonetheless, also during ongoing slow wave activity, we can identify components which likely resemble resting state networks, which seems reasonable, given that these networks are discussed to mirror basic functional connectivity.

Conventional model-based analysis for fMRI using the general linear model (GLM) solely identifies individual voxel significantly correlated with the chosen model timecourse. Data-driven, or model-free analysis as ICA, do not require a priori knowledge about activation patterns and furthermore consider interactions between voxel, as voxel are not considered individually. But it has to be noted that especially the ICA approach bears several significant drawbacks. For example this analysis can produce a large number of noise components. We used the minimum description length criterion (*Li et al., 2007*) to estimate the number of components, but also alternative techniques, as e.g. Bootstrap Stability Analysis (*Majeed and Avison, 2014*) are available. Still, this approach has to deal with the trade-off between extracting enough components to actually reveal all signals of interest and adding up spurious noise components. We used ICA to extract additional components which would be meaningful for our data, but were nevertheless able to identify the pan-cortical component of interest, as only this component showed high correlation to the timecourse revealed by the locally measured slow calcium wave signal. We furthermore used a seed-based approach to confirm our event-related GLM-based results.

## Translational perspective of using slow wave activity as a regressor for BOLD fMRI

Ongoing changes in network activity can contribute to the variability of experimental results and may appear as noise if they are disregarded in the statistical modeling of neural signals (*McGinley et al., 2015b*). A direct readout of neurophysiological measures may also be beneficial to monitor and control for homogenous network activity states during BOLD measurements, as depth of anesthesia may vary depending on many physiological factors, as e.g. cortical temperature (*Reig et al., 2010*; *Schwalm and Easton, 2016*; *Sheroziya and Timofeev, 2015*). From this perspective it is of high importance to relate local network activity to the global BOLD response, especially regarding the potential translational value of such studies.

Including ongoing signals revealed by local neurophysiological readouts as an independent variable, i.e. covariate, regressor, or entirely new factor depending on the type of analysis, could explain a considerable amount of variance in neuronal responses, leading to more reliable readouts, especially in global measures as fMRI.

We investigated the functional recruitment of brain areas by slow waves based on the reasoning that, if there is a global participation of cortical areas in slow wave rhythmicity, the activity of the cellular population as a whole should produce a macroscopic signal detectable by functional neuroimaging (*Doeller et al., 2010*). This study illustrates the power of combining cellular measures of neural activity with fMRI in systems neuroscience.

Our study directly relates a neurophysiological event – the calcium slow wave – to its potentially present BOLD correlate by employing a specific type of fMRI analysis (event-related), a straightforward and easy-to-apply approach as it is an inbuilt type of analysis of many standard fMRI analysis toolboxes. Results acquired with this method are in agreement with model-free and seed-based analysis approaches revealing a similar extension of network activity spanning almost the entire neocortex, also during ongoing slow wave activity.

Our results provide a framework for further analysis of slow wave brain activity within fMRI experiments in anesthetized rodents or for EEG/fMRI studies at a translational level (*Tagliazucchi et al., 2012*), since spatio-temporal dynamics and behavioral correlates of slow waves in humans and animal models are highly preserved (*Buzsáki et al., 2013*). For example, there is growing evidence of impaired slow oscillations in models of Alzheimer disease (*Busche et al., 2015*; *Menkes-Caspi et al., 2015*) and resting-state functional connectivity in Alzheimer patients (*Zhou et al., 2015*). Such studies could refine slow wave activity as potential disease marker (*Tagliazucchi and Laufs, 2014*) or specifically target impaired slow oscillations and their potential restoration. Even though calcium measures provide advantages in terms of spatial specificity (*Kajikawa and Schroeder, 2011*) and lack signal distortions by the varying magnetic fields of the scanner in comparison to electric population recordings, our event-related approach could also be applied for combined EEG-fMRI measures feasible in human experiments.

## Materials and methods

### Animals

Experiments were performed on 36 adult female Fisher rats (>12 weeks old, 160–180 g), of which optic fiber implantation in the cortex was performed in five animals (C1-C5) and optic fiber implantations in the thalamus in three animals (T1, T2, T3). Dual optic fiber implantation had been performed in two animals (DF1, DF2). LFP recordings were carried out in three animals (LFP1-LFP3) and resting state data acquisition without optic fiber implantation in 10 animals (M1-M10). Immunohistochemistry was done in four animals (H1-H4) and quantifications (amplitude, duration, rise time and probability of induction) of calcium responses and for the two different indicators were performed respectively in 8 (Q1-Q8) and in 2 (C3 and Q9) animals. Animals were housed under a 12 hr light–dark cycle and provided with food and water ad libitum. Animal husbandry and experimental manipulation were carried out according to animal welfare guidelines of the Westfalian Wilhelms-University Münster and the Johannes Gutenberg-University Mainz and were approved by the Landesamt für Natur-, Verbraucher- und Umweltschutz Nordrhein-Westfalen, Recklinghausen, Germany, and the Landesuntersuchungsamt Rheinland-Pfalz, Koblenz, Germany, respectively.

### Dye or virus injections and fiber placement

For surgical procedures including staining with fluorescent calcium indicator and optic fiber implantation, rats were anesthetized with isoflurane (maintenance 2–3%, Forene, Abbott, Wiesbaden, Germany), treated with subcutaneous injection of 1 mg/kg Metacam for analgesia, placed on a warming pad (37° C), and fixed in a stereotactic frame with ear and bite bars.

The skull was exposed, dried from blood and fluids, and leveled for precise stereotactic injections. Under a dissection microscope a small craniotomy was performed with a dental drill (Ultimate XL-F, NSK, Trier Germany, and VS1/4HP/005, Meisinger, Neuss, Germany). The fluorescent calcium sensitive dye Oregon Green 488 BAPTA-1 (OGB-1, Invitrogen, Life Technologies, Carlsbad, CA, USA) was prepared as described previously (*Garaschuk et al., 2006*), filtered, and injected into primary somatosensory cortex front limb area (S1FL; 0 mm AP,+3.5 mm ML, −0.5,–0.7 and 0.9 mm DV; n = 4 animals) or into the posterior thalamic nucleus (POm; three animals) at AP −3.3 mm, ML +1 mm, 12°, dorso-ventral 4.6, 4.9, 5.2 mm according to stereotactic coordinates (*Paxinos and Watson, 2006*). For co-staining of astrocytes, 10 mM of sulforhodamine 101 (SR101) was added to the OGB-1 solution in two animals (*Nimmerjahn et al., 2004*). Dye solution was delivered using a glass micropipette with an outer tip diameter of 45 µm and an inner diameter of 15 µm connected to a 10 mL syringe. Approximately 0.3 µL of the solution were slowly released at each depth by gentle manual pressure. After injection, the pipette was held in place for 2 min before slowly retracting it from the tissue. Alternatively, the genetically encoded calcium indicator GCaMP6f (UPenn Vector Core, PA, USA) was expressed via viral gene delivery four weeks prior to calcium/fMRI measurements. two animals received injections of approximately 1.0 µL AAV1.Syn.GCaMP6f.WPRE.SV40 in S1FL (0.0 mm AP, +3.0 mm ML, −1.2 mm DV) with an 35° injection angle. Two animals received approximately 1.0 µL AAV1.CamKII.GCamP6f.WPRE.SV40 at AP +0.2, ML +2.4, DV −1.6 with an 34° injection angle. A craniotomy (AP +0.2, ML +3.3) was performed on the day of the experiment. For the dual-fiber experiments a second craniotomy was performed at AP −1.8, ML +2.3 and approximately 0.3 µL OGB-1 was injected as described above.

After removing the cladding from the tip, an optic fiber was inserted perpendicular to the dura above the OGB-1 stained or GCaMP6f expressing region, typically at −300 µm (cortex) and −4.4 mm (POm), respectively and fixed to the skull with UV glue (Polytec, PT GmbH, Waldbrunn, Germany). The amount of glue for holding the fiber in place was kept at a necessary minimum to reduce MR image distortions.

After experiments rats were transcardially perfused with 4% paraformaldehyde (PFA) under deep isoflurane anesthesia. Coronal sections with 1000 µm slice thickness of acute brain slices were prepared from OGB-1/SR101 animals. For GCaMP6f animals 20 µm microtome slices (Leica CM 1850, Leica microsystems) of brains fixed overnight in 4% PFA, 30% sucrose, and embedded in Tissue Tec (Sakura Finetec Europe, NL) were prepared. Precise location of OGB-1 injections and GCaMP6f expressions were validated under a fluorescence microscope.

## Immunofluorescence

Coronal brain sections of 70 µm thickness of previously perfused animals were prepared using a vibratome (Leica, Wetzlar, Germany). For permeabilization/blocking, slices were incubated with 0.1% Triton X-100% and 5% normal donkey serum (Invitrogen, Life Technologies, Carlsbad, CA) in phosphate buffer solution for 90 min. Slices were incubated with rabbit anti-GAD 65/67 (1:200, Swant, Marly, Switzerland) or rabbit anti-CamKII (1:200, Epitomics, Burlingame, CA) at 4° C overnight. On the next day, slices were incubated with the secondary antibody Cy-3 donkey anti-rabbit (1:500, Jackson Immuno Research, West Grove, PA). Slices were mounted using antiquenching Vectashield (Vector Laboratories, Burlingame, CA), and confocal imaging was conducted using a Leica SP8 (Leica, Wetzlar, Germany) confocal microscope.

## Optical recordings

A custom-built setup was used for optic-fiber based calcium recordings (*Schmid et al., 2017*; *Schmid et al., 2016*). The light for excitation of the calcium indicators was delivered by a 20 mW solid state laser (Sapphire, Coherent, Dieburg, Germany) with a wavelength of 488 nm. To control laser power, an acousto-optic modulator (AOM 3080–125, Crystal Technology, Palo Alto, CA) was used. After deflection by a dichroic mirror, the beam was focused by means of a collimator into one end of a multimode fiber (Thorlabs, Grünberg, Germany) with a diameter of 200 µm and a numerical aperture of 0.48. The emitted fluorescent light was guided back through the same fiber and focused on a photodetector containing an avalanche photodiode (LCSA500-01, Lasercomponents GmbH, Olching, Germany) with an aperture of 0.5 mm. The recorded fluorescence signals were digitized with a sampling frequency of 2 kHz using a multifunction I/O device (PCI 6259, National Instruments, Austin, TX) and custom-written LabVIEW-based software (LabVIEW, National Instruments). Calcium data preprocessing was performed with Igor analysis software (WaveMetrics, Portland, OR; RRID: SCR_000325). All traces represent relative changes in fluorescence intensity ($\Delta f/f$) and were low-pass filtered before subsequent event detection. Rise time, duration and amplitude of calcium transients were assessed and statistically compared in animals measured upon OGB-1 injections (n = 8 animals) as well as in GCaMP6f transduced rats (n = 2 animals). Rise time was defined as time between onset of the transients and 50% of maximum intensity peak. Duration of the transients was defined as the time between the first and the last intensity value exceeding 50% of maximum intensity peak. Amplitude was determined as the intensity difference ($\Delta f$) from the baseline and the highest intensity value of the transients. For details see Source Data files 1 and 2 linked to *Figure 1*. Data were tested for normal distribution using the Lilliefors test, an assumption-free adaptation of the one-sample Kolmogorov-Smirnov test. In cases where normal distribution could be assumed (p>0.05), the parametric two-tailed Student's t-test was employed to compare means. Where normal distribution could not be verified, the non-parametric Wilcoxon rank-sum test for equal medians was used to test median differences. All quantifications are presented as mean ± SEM, unless stated otherwise. In cases of an unstable baseline or absence of characteristic signal dynamics of slow wave-associated calcium transients (*Grienberger et al., 2012*; *Stroh et al., 2013*) calcium data was excluded from analysis (n = 2 animals).

## Calcium slow wave detection

Data was read into Matlab (The Mathworks, Inc., Natick, MA; RRID:SCR_001622) and downsampled from 2 kHz to 1 kHz by averaging two adjacent values. The baseline of the calcium signal was determined and corrected by using the Matlab function *msbackadj* estimating the baseline within multiple shifted windows of 2500 datapoints width, regressing varying baseline values to the window's datapoints using a spline approximation, then adjusting the baseline in the peak range of the calcium signal. This method provides a correction for linear and low-frequency drifts while avoiding contributions of signal-of-interest events. We employed a procedure based on exponential moving average (EMA) filters, originally established to identify slow wave activity in electrophysiological recordings (*Seamari et al., 2007*). According to this algorithm, to detect slow waves, an EMA filter with a window of 25 ms was applied. Then, the datapoints of the filtered signal exceeding or falling below a threshold were used as preliminary transition points between slow calcium waves and silent periods.

To determine this threshold, amplitudes of all datapoints were sorted into a histogram. Onsets and durations for each threshold (in percent of maximum) were derived and the resulting numbers of active voxel of SPM-evaluations using these onsets and durations showed an inflection point at 70%, therefore the threshold was set to 70% for further evaluations (*Figure 2—figure supplement 2*).

Onsets of slow calcium waves were defined as signal timepoints exceeding the threshold (70%), termination of calcium slow waves was defined as signal timepoints dropping below 50% of the threshold value. As EMAs are optimized to detect onsets of events but provide less precision regarding event terminations, these were corrected with regard to the noise level. Further, the detected activity was post-processed in the following order: (1) calcium waves separated by a time-interval below 100 ms were interpreted as one wave; (2) calcium waves with a duration of less than 600 ms were discarded; (3) activity not reaching 90% of the histogram-based cumulative signal intensity was discarded.

Frequencies of calcium slow wave events were evaluated by separating the measurements into bins comprising of 100 s duration of measurement time. For each bin, the number of detected events was counted and the frequency of these events was recalculated in Hz.

Onset and duration time points of calcium slow waves were extracted and subsequently used for the fMRI analysis described below.

## fMRI Data acquisition

fMRI data acquisition under isoflurane (1.2–1.8% in 80% air and 20% oxygen) was performed on a 9.4 T small animal imaging system with a 0.7 T/m gradient system (Biospec 94/20, Bruker Biospin GmbH, Ettlingen, Germany). The optic fiber-implanted animals were mounted on a heated MRI cradle. The skull was covered with a 1–2 mm thick layer of dental alginate (Weiton, Johannes Weithas dental-Kunststoffe, Lütjenburg, Germany) or 1% agar, to reduce susceptibility artifacts which may occur at the tissue air boundary. The fiber was guided through a lead-through in the RF surface head coil and local shimming was applied (shimming volume 0.6–2.2 cm$^3$, first order Mapshim, Bruker). Anatomical images were acquired with a T2-weighted 2D RARE sequence, TR/TE 2000/12.7 ms, RARE factor 8, 256 matrix, 110 × 100 μm$^2$ spatial resolution and slice thickness 1.2 mm, 6–16 contiguous slices prior to fMRI scans to check for tissue damage and validate the fiber's location. For BOLD fMRI measurements, T2*-weighted images were acquired with a single-shot gradient echo EPI sequence with TR = 1 s (1.5 s for animals M1-M10), TE = 18 ms (14 ms for animals M1-M10), and FA 65°, 350 × 325 μm$^2$ spatial resolution (320 × 290 μm$^2$ for animals M1-M10), slice thickness 1.2 mm (0.8 mm for animals M1-M10), 9–16 continuous slices (34 for animals M1-M10), 2400 (n = 3), 1800 (n = 5) or 1200 (n = 10) acquisitions resulted in a scan time of 40, 30 or 20 min, respectively.

Experiments were performed at least 1 hr after OGB-1 injection (*Stosiek et al., 2003*) under iso-flurane anesthesia (1.1–1.8%) inducing slow wave activity (*Grienberger et al., 2012*; *Stroh et al., 2013*). Body temperature and respiration rate were routinely monitored during experiments.

fMRI data acquisition under medetomidine (n = 5) was performed under the same parameters as under isoflurane. For the recordings animals were anesthetized with 1.5% isoflurane and mounted in on a heated MRI cradle in the scanner. Anatomical images with the same parameters as the previous experiment were acquired. During the acquisition of the T2 images a bolus of 0.04 mg/kg of mede-tomidine was administered to the animals. Five minutes later, isoflurane anesthesia was turned off and medetomidine 0.08 mg/kg/h was perfused until the end of the experiment.

## fMRI analysis

### GLM analysis

GLM fMRI data processing was performed using Matlab software environment and SPM12 (Functional Imaging Laboratory, Wellcome Trust Centre for Neuroimaging, UK London; RRID:SCR_007037). Preprocessing steps included realignment, reslicing and smoothing of the images, using asecond degree B-Spline interpolation method for realignment and a fourth degree B-Spline for reslicing. Serial autocorrelation was performed with an autoregressive AR(1) model during classical (ReML) parameter estimation, datasets were smoothed using a 0.5 mm FWHM Gaussian Kernel and highpass filtered with a cutoff frequency of 1/128 s. The initial five images were discarded to only include data for which signal had reached steady state.

To approximate the underlying timecourse of hemodynamic responses (HRs) in the raw data, the hemodynamic response function (HRF) was derived using in a different sub-set of animals a model-free approach based on a finite impulse response filter (*Glover, 1999*). The FIR set consists of a number of successive post-event time bins and effectively averages the BOLD response at each post-event time. Note that the statistical efficiency of this approach increases with decreasing and jittered onset intervals (*Dale, 1999*). To detect the HR upon slow calcium waves without making a priori assumptions about the shape of the HR, estimates were computed at 40 time points according to the TR with an interval of 1 s, ranging from 20 s prior to the start of an onset to 20 s after the start (*Agam et al., 2014*; *Masterton et al., 2013*). The actual response upon a slow calcium wave is considered as the resulting shape of the timecourse starting at 0 s up to 20 s (*Figure 2E* shows the derived timecourses 5 s prior onset to 20 s after onset).

The onsets and durations of spontaneous calcium waves were binarized to define an array of input functions: slow calcium wave events ('1') and silent periods ('0'). These 'slow wave vectors' define a condition for an event-related SPM analysis (*Friston et al., 1998*) and were convolved with a basis set to yield regressors subsequently being correlated with the raw data (see below).

For the FIR basis function, window size was set to 40 s. According to TR = 1 s a total number of 40 bins was used, leading to 40 beta maps (*Henson, 2001*). Therefore, each beta map corresponded to 1 s within the timecourse, while the interval between beta maps was 1 s and the total timecourse was 40 s. F-tests including all 40 beta maps were calculated with a contrast threshold of $p<0.05$. Starting from the slice containing most active voxel in the resulting F-maps, we extracted the 10 voxel with highest activation and did so for the previous and the following slice to identify in total 30 voxel of highest F-value in three slices for the extraction of signal timecourses. Voxel in the sinus vein were manually masked to exclude them from the selection of activated voxel. All identified voxel were checked for localization. The HRs of the identified voxel were extracted using a custom-written Matlab script exporting the value of each voxel for each beta map. The resulting 30 HRs were averaged to derive the mean BOLD response upon a spontaneous slow calcium wave. For analysis of each animal, extracted timecourses of either 2 or 3 other animals were used to avoid circular analysis (*Kriegeskorte et al., 2009*), for this procedure HRs from animal C4 were not included because of minor signal loss due to susceptibility artefacts (*Figure 3—figure supplement 2*). The extracted HRs were averaged and used as regressors by manually convolving the extracted, averaged timecourses, considering data points from 0 s (onset) until 15 s when the timecourses reached baseline level, with the slow wave vectors (event arrays). A t-test was performed using a contrast threshold of $p<0.05$ (FWE) for the resulting activation maps. For an overview of the analysis work flow see *Figure 2G*.

To confirm that resulting activation maps were independent from the considered averaged extracted HR used as basis function, a cross-validation was performed. Therefore, the event array of the animal to be analyzed was convolved with the extracted HR from one animal and the procedure was reiterated convolving the event-train with the extracted HR of another animal. Statistical threshold was set to $p=0.05$ (FWE) and T-values are reported and compared.

To control for specificity of results, we exchanged calcium event arrays between animals and used onset and duration time points to create a regressor for analyzing corresponding fMRI data. In addition, to test results upon arrays containing the same number of events and the same distribution of event durations and inter-event intervals but with differential event timing, we mirrored the originally detected events for each experiment in time and used them likewise for the fMRI analysis.

For quantitative comparisons of regional activity (*Figure 3E*) mean images of the functional scans, F-maps, and 20 beta-maps were loaded into ImageJ and slices on the level of somatosensory (S1) and visual cortex (V1) were identified according to anatomical landmarks. Ipsi- and contralateral ROIs containing a constant number of 23 pixels were drawn for S1 and V1 respectively with respect to anatomical structures in all regions devoid of susceptibility artefacts. The mean F- and beta-values of all 20 beta-maps were calculated in these ROIs. F-values were tested for normal distribution (Lilliefors test) and subsequently compared for equality of the means by Student's t-test. Beta values were averaged to receive an average HR of every cortex region. The two resulting HRs were compared using a Wilcoxon rank-sum test.

To compare the influences of the FIR-HRF model used for the analysis to the classical canonical HRF model, we reiterated the analysis using the classical SPM procedure with the standard canonical HRF which is modeled by two gamma functions. We compared T-values derived from the canonical

HRF and the FIR-HRF in ipsi- and contralateral ROIs as described above. The mean values of these ROIs for each method in the corresponding T-maps were tested for normal distribution (Lilliefors test) and compared using a paired t-test or Wilcoxon rank-sum test. ROIs were drawn based on the mean functional images on S1FL, ipsi- and contralateral to the side of optic fiber recordings. The averaged HRs upon slow calcium waves were analyzed in terms of latency in respect to the preceding calcium wave, time-to-peak (TTP), normalized peak amplitude ($\Delta$SA) and half-maximum-duration (HMD), for every voxel in all ROIs. For this purpose data were smoothed by a moving average filter (average of 5 time points, i.e. 5 s) and normalized according to the baseline signal defined as the mean signal intensity of 5 s preceding the calcium wave onset. Timecourses were averaged for each event and each voxel within the ROI. The resulting timecourse was fitted by a gamma variate-function (*Yu et al., 2016*), yielding a correlation coefficient of $r^2 = 0.96 \pm 0.02$ (n = 4 experiments). The onset of HRs was defined as fit parameter $t_0$ and TTP, $\Delta$SA and HMD were calculated.

For the calcium recordings-informed correlation analysis, baseline was corrected as described above, but with a window width of 100 s, and averaged HRs of selected ROIs were correlated with the signal over the entire timecourse from every voxel in the slice. For this purpose, HRs upon slow calcium waves as defined by the event array were averaged over selected ROIs and used as a reference for the HRs derived from the voxel-by-voxel analysis. To derive regions of highest correlations, voxel values with r > 0.8 were displayed, clusters of less than 70 contiguous voxel were removed and the resulting region was superimposed onto the mean functional data set. For every voxel of this region, a parameter map of TTP was derived from the voxel-by-voxel data as the duration between the slow wave onset and timepoint of maximum signal value.

## Independent component analysis

For the independent component analysis the freely available software 'Group ICA of fMRI Toolbox (GIFT)", version 4.0b (http://mialab.mrn.org/software/gift/; RRID:SCR_001953) was used. Mask-calculation using the first image was employed and time-series were linearly detrended and converted to z-scores. For all other settings the following default parameters were used. For back-reconstruction, the 'GICA'-algorithm (*Calhoun et al., 2001*) has been used and all results have been scaled by their z-scores. We used the Infomax-algorithm with regular stability analysis and no autofill was performed. The options for the Infomax-algorithm were as follows: Block: 119, Stop: 1e-006, Weight: 0, Lrate: 0.01082, MaxSteps: 512, Anneal: 0.9, Annealdeg: 60, Momentum: 0, Extended: 0. Images for the ICA were motion corrected and smoothed as described above for each animal. To be able to perform a group analysis, the functional data was co-registered to the anatomical data and the software DARTEL (*Ashburner, 2007*) was used to normalize the anatomical data as described (*Kronfeld et al., 2015*; *Sawiak et al., 2013*). The resulting DARTEL 'flow fields', which are containing the normalization parameters, were applied to the functional data to normalize all images to the same space consisting of a $96 \times 54 \times 58$ voxel matrix with $0.3$ mm$^3$ isovoxel resolution. BOLD activation patterns revealed by ICs are shown on 21 of the 96 resulting reconstructed coronal slices with a slice distance of 0.6 mm.

For the group-level-ICA (n = 4) the mean minimum description-length (MDL)-criterion (*Li et al., 2007*) was used which revealed 37 ICs to be calculated. For group ICA standard PCA was performed with two data-reduction/pre-whitening steps: the number of principal components (PC) is first calculated from all data defined as 1.5 times the number of ICs, resulting in 56 PCs for the pre-whitening step. In the second step the concatenated datasets of all subjects is reduced to the number of extracted ICs (37). Results were inspected visually and ICs that showed only activity outside of the brain were discarded. The remaining 23 ICs were displayed (group analysis, 23 mean ICs of n = 4 animals, and on single subjects basis underlying the group analysis), sorted by the mean power of their frequency spectrum derived from a Fast Fourier transform (FFT) of their timecourses, in the range up to 0.1 Hz.

For ICAs performed on single subject basis (n = 15; animals C1-C5, T1 and T2, M1-M6, M8 and M9) the number of ICs to calculate for each animal was set to 20 for assuring the same number of components for each animal and representing a similar number of ICs as identified in the group analysis as detailed above. For single-subject ICA only one data-reduction-/pre-whitening-step is performed with the number of PCs equaling the number of ICs. Spatial allocation of ICs was inspected

visually. The ICs timecourses were related to the regressors used in the SPM evaluation by cross-correlation analysis. FFTs were calculated for IC timecourses to derive their power spectra.

For regressing out the calcium slow wave vector from the BOLD timecourse, we used the resting state fMRI Data Analysis Toolkit (*Song et al., 2011*). The function 'rest_RegressOutCovariates' was used to eliminate the signal part of the SPM-regressors from the BOLD-signal in every voxel over time. A single-subject ICA as described above was performed prior and after this procedure.

### Seed-based analysis

To perform seed-based analysis, fMRI data was preprocessed as previously described for the GLM analysis. The somatosensory ROIs were defined on subject-by-subject basis, based on active voxel upon a forepaw stimulation localizer. The mean signal in these functionally defined ROIs was then subsequently extracted in the resting conditions. For the hippocampal formation each subject's functional data was manually aligned to a template obtained from Valdés-Hernández *et al.* (*Valdés-Hernández et al., 2011*) using Brain Voyager QX (vs. 2.8.4, Brain innovation, Maastricht, The Netherlands). According to the resulting atlas for each animal, the mean signal of the hippocampal ROI was extracted.

These mean signals were correlated with the signal of each cortical voxel. To perform this correlation, we used a cortical mask based on the previously employed atlas template (*Valdés-Hernández et al., 2011*). Voxel and cluster were then quantified from the correlation maps obtained from this step. Significance threshold for r-values was set to 0.5. Finally, for the slow wave fMRI data, we correlated the mean signal of the ROIs for each subject with the timecourse of the pan-cortical component obtained from the previously calculated ICAs.

## Electrophysiological recordings

In a series of control experiments (n = 3) simultaneous local field potential (LFP) recordings and calcium measurements (as described above) were performed outside the MR scanner. A pipette with a tip resistance of 0.2 MΩ filled with Phosphate Buffered Saline (Sigma, Munich, Germany) was inserted through a second craniotomy at a depth of 300 μm, lateral to the optic fiber insertion site. A reference electrode was inserted into the cerebellum, 1 mm posterior from Lambda and signals were amplified using an extracellular amplifier (EXT-02F/2, npi Electronics, Tamm, Germany). Signals were filtered at 300 Hz (low pass), digitized at 2 kHz and recorded together with the optical signals using LabView (RRID:SCR_014325).

## Acknowledgements

This work was supported by the DFG (Fa474/5, SFB 1080, SFB 1193, SPP 1665); the Focus Program translational Neurosciences (ftn), the Interdisciplinary Center for Clinical Research Münster (Fa3/1603, PIX) and the Excellence Cluster Cells in Motion (DFG EXEC 1003). We thank Nina Nagelmann for excellent technical assistance; Lara Hamzehpour for helping with data collection and analysis; Nathalie Just, Eduardo Rosales Jubal and Kenneth Yuen for advice.

## Additional information

### Funding

| Funder | Grant reference number | Author |
|---|---|---|
| Deutsche Forschungsgemeinschaft | Fa474/5 | Cornelius Faber |
| Deutsche Forschungsgemeinschaft | SFB 1080 | Albrecht Stroh |
| Deutsche Forschungsgemeinschaft | SFB 1193 | Albrecht Stroh |
| Deutsche Forschungsgemeinschaft | SPP 1665 | Albrecht Stroh |

| | | |
|---|---|---|
| Focus Program Translational Neurosciences | | Miriam Schwalm<br>Albrecht Stroh |
| Interdisciplinary Center for Clinical Research Münster | Fa3/1603,PIX | Cornelius Faber |
| Excellence Cluster Cells in Motion | EXEC 1003 | Cornelius Faber |

The funders had no role in study design, data collection and interpretation, or the decision to submit the work for publication.

## Author contributions

Miriam Schwalm, Conceptualization, Formal analysis, Investigation, Visualization, Writing—original draft, Writing—review and editing; Florian Schmid, Data curation, Formal analysis, Visualization, Methodology, Writing—review and editing; Lydia Wachsmuth, Data curation, Methodology, Writing—review and editing; Hendrik Backhaus, Felipe Aedo Jury, Pierre-Hugues Prouvot, Data curation, Formal analysis, Visualization; Andrea Kronfeld, Formal analysis, Visualization; Consuelo Fois, Franziska Albers, Timo van Alst, Data curation; Cornelius Faber, Conceptualization, Funding acquisition, Project administration, Writing—review and editing, Equally contributing last author; Albrecht Stroh, Conceptualization, Supervision, Funding acquisition, Writing—original draft, Project administration, Writing—review and editing

## Author ORCIDs

Miriam Schwalm http://orcid.org/0000-0003-4162-2298
Cornelius Faber http://orcid.org/0000-0001-7683-7710
Albrecht Stroh http://orcid.org/0000-0001-9410-4086

## Ethics

Animal experimentation: Animal husbandry and experimental manipulation were carried out according to animal welfare guidelines of the Westfalian Wilhelms-University and the Johannes Gutenberg-University Mainz and were approved by the Landesamt für Natur-, Verbraucher- und Umweltschutz Nordrhein-Westfalen, Recklinghausen, Germany (animal protocol number: 84-02.04.2015.A427), and the Landesuntersuchungsamt Rheinland-Pfalz, Koblenz, Germany (animal protocol number: G 14-1-040). All surgery was performed under deep isoflurane anesthesia.

## Decision letter and Author response

Decision letter https://doi.org/10.7554/eLife.27602.029
Author response https://doi.org/10.7554/eLife.27602.030

## Additional files

### Supplementary files

• Transparent reporting form
DOI: https://doi.org/10.7554/eLife.27602.028

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
