## [Decision Letter]

[Editors’ note: a previous version of this study was rejected after peer review, but the authors submitted for reconsideration. The first decision letter after peer review is shown below.]

Thank you for submitting your work entitled "Locally recorded slow calcium waves are associated with cortex–wide BOLD fMRI activity" for consideration by *eLife*. Your article has been reviewed by four peer reviewers, one of whom, Jan–Marino Ramirez is a member of our Board of Reviewing Editors and the evaluation has been overseen a Senior Editor. The following individual involved in review of your submission has agreed to reveal their identity: Sylvain Crochet (Reviewer #3).

Our decision has been reached after consultation between the reviewers. Based on these discussions and the individual reviews below, we regret to inform you that your work will not be considered further for publication in *eLife*.

As you can see from the critiques the reviewers saw the technical advances of your study, but were concerned by the biological significance. It is indeed very important that you demonstrated that the waves extend globally, and that you can link the calcium imaging with the fMRI signal. However, the use of anesthesia is a major caveat. The use of anesthetics may explain the global spread, which might be different from natural SWS. While we understand that the lack of thalamic involvement may be merely a technical issue, one reviewer felt that this could also be due to the anesthesia. We hope you can address these comments in a new manuscript that carefully discusses these caveats, emphasizes the technical advances and the importance of the biological significance of your finding.

Also, please carefully explain the lack of thalamic activity, the rationale for using these particular anesthetics and the caveats associated with this.

Reviewer #1:

Default state functional MRI has become an increasingly important tool to explore various neurological and psychiatric conditions. This approach, which was pioneered by Dr. Marcus Raichle, can be used to characterize addiction, demonstrate developmental changes in cortical connectivity between different brain regions and identify developmental delays. The importance of this approach is documented by an increasing number of publications in high profile Journals. A major limitation of this approach is that the relationship between the BOLD signal and neuronal activity is complicated and there are those that believe this signal reflects primarily hemodynamic changes rather then actual changes in neuronal activities. One of the main technical limitations is that the use of MRI technology is difficult to combine with conventional electrophysiological techniques, because the magnetic fields are disturbed by the metallic nature of wires. The major strength of this study is the use of optical techniques that allows the combined characterization of neuronal signals and the wide-range Bold signal using fMRI approaches. The present study has also another strength in that it characterizes these activities in "small" animals. This has the major advantage that wide range optical signals can be characterized.

This study is clearly of great interest for a general readership: the demonstration that locally recorded calcium waves correlate with BOLD signals spanning distant cortical states is important since understanding the neuronal basis of cortical states is critical for understanding sensory processing, the generation of sleep and wake states and the role of memory consolidation, just to name a few examples. The study can relate slow wave onsets to fMRI fluctuations, which has not been possible before. These slow waves have been well characterized by other neurophysiological studies, which closes the loop between a well-characterized neuronal activity and the macroscopic fMRI signal. The study therefore strongly suggests that the cortex-wide BOLD signal is a property of slow-wave neuronal activity. The study seems to be well powered and carefully conducted. The introduction and discussion is very interesting to read and relates the finding in this rodent model to various general neurobiological phenomena. Thus, I don't have major concerns.

Reviewer #2:

These authors combined fMRI with a fiber optic–mediated calcium recording method to study the slow calcium waves in the rodent brain. It is an excellent combination of techniques to study brain function at multiple scales. The main effort of this work is to decipher the potential linkage of slow calcium waves with the resting state fMRI signal of the anesthetized rodent brain, which is an important and novel direction for small animal fMRI studies. Besides the technical advancement, more discussion could be provided to better clarify the biological significance and image processing strategy. The reviewer has a few concerns needed to be clarified by the authors:

1) The calcium signal was recorded in rats anesthetized with isoflurane. The trace of local field potential (Figure 1) mimics the burst suppression events. It will be better to discuss the states of the anesthetized animals of this work in comparison to the literature using similar drugs (or other anesthetics) to study slow wave activity.

2) The procedure to estimate hemodynamic response function (HRF) should be carefully described. One puzzling issue is the time course of the HRF, which showed very fast onset time in comparison to the existing models (Figure 2). The authors stated "Time course of averaged HRs exhibited a sharp rise 0.7 s {plus minus} 0.4 s after the onset of a slow calcium wave." What does the "sharp rise" mean (equal to onset time?). If it is the case, please discuss the early onset of the estimated HRF by referring to the existing literature on this matter. If the authors presented the time courses of simultaneously acquired calcium and fMRI signal, would they have observed any phase shift? It will be better to provide a cross correlation curve to specify the temporal features of the two kinds of signals.

3) One of the key conclusions of this work is that the functional significance of the "slow calcium waves" specifies global correlation of the whole cortex. The authors suggested that the slow calcium waves were related to the slow wave activity as discussed by the authors. It will be better to explain the underlying rationale that no subcortical regions detected in the correlation maps.

Reviewer #3:

In this manuscript Schwalm et al., have developed an innovative approach to study global patterns of activation brain–wide in relation to local slow–wave activity, combining local calcium imaging and fMRI in anesthetized rats. Using different analytical approaches, the authors show that cortical slow–waves detected locally with calcium imaging, are correlated to a global, cortex–wide activation detected with BOLD fMRI. This is an elegant and well-conducted study and I have little concern about the methodology – but I must say that I have little expertise in fMRI.

My main issue with this study concerns the limitation of the interpretation of the results that need to be addressed in the discussion. Indeed, the authors make a strong statement about highly coherent slow–waves activity across the entire cortex based on their results obtained in deeply anesthetized rats. Anesthesia is not SWS. Although anesthetics have been used extensively to characterize slow–wave activity and determine its cellular mechanisms – because of the apparent similarity of the brain activity under some anesthetics and natural SWS – many studies also point directly or indirectly to critical differences between anesthetized states and natural SWS:

1) Down– or Silent–states are shorter during SWS compared to anesthesia and last rarely longer than 1s (Steriade et al., 2001; Timofeev et al., 2001; Mahon et al., 2006; Chauvette et al., 2011). In this paper, the authors report patterns of activity with very long Silent–states (~ 5 s or more) between consecutive slow–waves, which indicates very deep levels of anesthesia. Although it is easy to understand the rational for maintaining such depressed cortical activity to provide enough temporal separation of single slow–wave events in this study, the relevance of the data acquire in such anesthetized state to natural SWS is questionable.

2) While under anesthesia slow–wave activity is highly synchronized across cortical areas, the long–range cortico–cortical coherence during natural SWS is much lower (Chauvette et al., 2011; Busche et al.,), probably due to the fact that slow–waves often occur very locally (Nir et al., 2011). As a result, many studies point to a decreased long–range functional connectivity during SWS compared to wakefulness (Olcese et al., 2016; Tagliazucchi et al., 2013). Therefore, the conclusion that slow–waves are highly coherent across cortex might only be true under anesthesia.

3) SWS seems to promote, or even to be critical, for memory formation, whereas, to the best of my knowledge, anesthesia has a rather negative impact on memory formation.

The authors should address these points in their discussion and moderate their conclusions.

Reviewer #4:

What is the question asked by the manuscript? What is the answer provided? The manuscript appears as an elegant exercise to correlate slow brain activity with BOLD signal, I can't evaluate whether this is new. It doesn't seem new based on the references. There is nothing new in terms of biological data, the widespread and synchronized nature of slow oscillations is well known since the original description of the slow oscillation by Steriade et al., (1993). In those papers, also the synchrony with thalamus was established and then confirmed repeatedly, it is surprising that the thalamus shows no signal, slow oscillation in synchrony with cortex have also been shown in basal ganglia, why is there no signal in basal ganglia?

Why are these called calcium waves? Does it make reference to propagation? Or are these just increments in calcium signal with weak periodicity? Please clarify and be consistent throughout text.

I don't understand the statement in the subsection “Cortical slow wave–associated population activity can be reliably detected by optic calcium recordings during fMRI scans”, or what its significance is. ("This local confinement allows for a clear attribution of slow wave activity to a spatially restricted region") Also, the statement about the LFP is incorrect, if the LFP is recorded in bipolar configuration it is as confined or more than the calcium signal. Only in monopolar configuration (with a reference elsewhere in muscle or bone) the statement of volume conduction is true. In the original description of the slow oscillations by Steriade et al., (1993), the recordings are bipolar between the surface and depth of a cortical column and the intracellular recordings were obtained from the immediate vicinity of the LFP. By doing cross correlation between recording sites from spatially confined regions at different distances is how global synchrony was demonstrated, such demonstration would have been impossible with monopolar LFPs which are contaminated by volume conduction.

Furthermore, the paper by Katzner et al., 2009 [cited?] reaches the opposite conclusion than Kajikawa and Schroeder (Cited in the manuscript) and states that the LFPs represent mostly local activity within 450 microns. The paper by Linden 2011, reaches yet another conclusion: that the area of tissue represented by the LFP depends on the degree of correlation of the underlying networks. Thus, the authors do a poor job here and this part of the manuscript should be rewritten.

The HR peak at 6.5 s contains at least 5 calcium waves, how is the HR related to an individual calcium wave? Calcium waves occur rhythmically and continuously, how is the effect of subsequent calcium waves eliminated from HR? What does the decay signal of HR represent? How can the HR signal decay many seconds after the calcium wave? Please explain.

The HR suggests that the entire brain is having increases and decreases of oxy/deoxy Hb, which are more than an order of magnitude longer than the underlying oscillations. How is this possible? What is driving the HR?

The finding that BOLD signal in the entire cortex is not new and confirms the early findings of Steriade et al., 1993. In the subsection “Calcium slow wave event–related analysis reveals cor tex–wide BOLD correlate”, the authors indicate that the data "suggest" but in reality the data "agrees" with previous descriptions.

Subsection “Calcium slow wave event–related analysis reveals cortex–wide BOLD correlate”: functional assemblies under isoflurane anesthesia? That sounds absurd. Widespread slow activity is the exact opposite of functional assemblies. The brain is anesthetized. Unless the authors refer to a functional state related with deep sleep? What would it be?

This manuscript should not be published or considered further without clear answers and clarifications to the above questions.

[Editors’ note: what now follows is the decision letter after the authors submitted for further consideration.]

Thank you for submitting your work entitled "Cortex–wide BOLD fMRI activity reflects locally–recorded slow oscillation–associated calcium waves" for consideration by *eLife*. Your article has been reviewed by three peer reviewers, one of whom, Jan–Marino Ramirez is a member of our Board of Reviewing Editors and the evaluation has been overseen a Senior Editor. The following individuals involved in review of your submission have agreed to reveal their identity: Sylvain Crochet (Reviewer #2).

Our decision has been reached after consultation between the reviewers. Based on these discussions and the individual reviews below, we regret to inform you that your work will not be considered further for publication in *eLife*.

This study is very innovative with the use of cutting–edge calcium imaging techniques and the combination with fMRI. This combination can potentially provide critical insights into the relationship between neuronal activity and the BOLD signal used for fMRI studies. Thus, the study is very timely, as the use of fMRI approaches is becoming an increasingly important tool in translational neuroscience. While the reviewers acknowledged the improvement following the careful revision of the manuscript and the addition of the de–synchronized state, they also raised several important methodological and conceptual concerns. As you can read from the reviews, a major issue was the design and analysis of the fMRI study, which limits the interpretation of your data. In addition, one reviewer was still concerned with regards to the use of anaesthesia as a method to characterize a natural brainstate.

Reviewer #1:

The authors have done a great job in revising the manuscript. They added the characterisation of a desynchronised state, and they showed slow oscillations in the thalamus, which was a major concern in the previous submission. The authors use light anaesthesia, which was a confounding issue in the first submission, but the first submission did not show the desynchronised state, which differs fundamentally from the more synchronised state that has similar characteristics of SWS.

The comparison with the BOLD signal is interesting and will contribute to our understanding of what neuronal activity actually underlies this signal.

*Reviewer #2:*

In this manuscript Schwalm et al., use an innovative approach to study global patterns of activation brain–wide in relation to "local" slow–wave activity, combining local calcium imaging and fMRI in anesthetized rats. Using different analytical approaches, the authors show that cortical slow–waves detected locally with calcium imaging, are correlated to a global, cortex–wide activation detected with BOLD fMRI.

In this revised version, the authors also compare the pattern of activation during slow–wave activity in deep anesthesia to a more desynchronized brain activity under sedation. They find that the cortex wide activation during slow–waves contrast with the more restricted activation during sedation similar to the default mode network. The authors also studied the BOLD correlate of thalamic slow–waves, finding again a cortex–wide activation but no thalamic activation. The authors argue that the absence of thalamic activation during slow–waves could result from technical limitation.

Although this revised version provides new and interesting data, the authors have not address my main concerns and I keep thinking that the relevance of this study is considerably limited by the use of deep level of anesthesia and technical limitation – in particular the very poor temporal resolution of fMRI. Hence, my comments for this revised version are very similar to my initial comments.

However, I suppose this study might be of interest for the field of fMRI.

Please refer to my previous comments (then as Reviewer #3), appended here for reference:

My main issue with this study concerns the limitation of the interpretation of the results that need to be addressed in the discussion. Indeed, the authors make a strong statement about highly coherent slow–waves activity across the entire cortex based on their results obtained in deeply anesthetized rats and with poor temporal resolution. Anesthesia is not SWS. Although anesthetics have been used extensively to characterize slow–wave activity and determine its cellular mechanisms – because of the apparent similarity of the brain activity under some anesthetics and natural SWS – many studies also point directly or indirectly to critical differences between anesthetized states and natural SWS:

1) Down– or Silent–states are shorter during SWS compared to anesthesia and last rarely longer than 1s (Steriade et al., 2001; Timofeev et al., 2001; Mahon et al., 2006; Chauvette et al., 2011). In this paper, the authors report patterns of activity with very long Silent–states (~ 5 s or more) between consecutive slow–waves, which indicates very deep levels of anesthesia. Although it is easy to understand the rational for maintaining such depressed cortical activity to provide enough temporal separation of single slow–wave events in this study, the relevance of the data acquire in such anesthetized state to natural SWS is questionable.

2) While under anesthesia slow–wave activity is highly synchronized across cortical areas, the long–range cortico–cortical coherence during natural SWS is much lower (Chauvette et al., 2011; Busche et al., 2015, Fernandez et al., 2016), probably due to the fact that slow–waves often occur very locally (Nir et al., 2011). As a result, many studies point to a decreased long–range functional connectivity during SWS compared to wakefulness (Olcese et al., 2016; Tagliazucchi et al., 2013). Again, many studies point to slow–wave events being rather local than global during natural slow–wave sleep. Therefore, the conclusion a cortex–wide activation during slow–waves events might only be true under anesthesia.

3) SWS seems to promote, or even to be critical, for memory formation, whereas, to the best of my knowledge, anesthesia has a rather negative impact on memory formation. In order to synchronize cortical activities across areas and efficiently drive plasticity, the precise timing – at milliseconds scale – of slow–waves is critical (Miyamoto et al.et al., 2016). Such time precision cannot be achieved with fMRI.

The authors should really address these points in their discussion and moderate their conclusions.

Reviewer #3:

Essentially, it seems to me the authors are trying to prove that there is cortex–wide BOLD activation in the rat brain that is driven by (or at least correlates to) activation in calcium slow–wave response by means of demonstrating that one can convolve calcium slow wave tracings with a FIR basis set and fit this to the BOLD data. However, FIR basis sets are complex alternatives to standard models of the HRF and while they allow a large amount of flexibility this comes at the price of bias–variance tradeoff, and they should be used within rigorous multivariate models. Generally, FIR basis sets are used in studies aiming at characterizing the shape of the HRF, and not for studies aiming to detect activations. This study aims at detecting activation. As well, there are a very small number of subjects, and many substantial questions in the methods used to process the BOLD, ranging from sequence used to pre–processing to use of ICA. It is not clear to me what the authors have actually "detected" in the BOLD – it may just be motion, or other artifact.

Specific Methods Concerns

I have substantial concerns with the methods described to acquire, pre–process and analyze the resting–state fMRI data. These include but are not limited to:

1) Sequence: a very short TR of 1 sec was used. This is unconventional and has disadvantages in the acquisition of resting–state (in rodents, or humans). Generally, a TR of at 2–3 secs is preferred.

2) Pre–processing pipeline: this is (thinly) described as being realignment, reslicing and smoothing. This raises the following concerns:

a) Coregistration and normalization do not appear to have been performed prior to smoothing. Realignment only performs within–subject motion correction. It does not perform between–subject motion correction. The authors describe a process (subsection “fMRI analysis”) where they take heavily processed maps and manually align them with anatomical images (presumably for purposes resembling coregistration) using ImageJ. This is of concern because (a) coregistration should be performed earlier in the pipeline, prior to smoothing (b) They are performing a manual process and not using an appropriate normalization algorithm and (c) Using a piece of software intended for the processing of visual (not neuroimaging) images. The authors should re–perform their pre–processing pipeline to incorporate coregistration and normalization prior to smoothing. Excellent choices are readily available in SPM12, which they are using.

b) Smoothing has the effect of boosting SNR. It is generally used in pipelines where masses of voxels are being considered in larger numbers of subjects. Here, a small number of voxels is being considered in a small number of subjects. How do we know the resulting 'signal' is not artificially boosted by the smoothing process?

c) The lack of normalization and use of manual methods in ImageJ introduces many concerns regarding anatomic standardization among the various parts of this complex pipeline. It is not clear to this reviewer how integrity is maintained in comparing one location vs another between processes and animals. For example, the sites of optical fiber recordings, ROI, etc. – why have conventional methods not been used to place all locations in MNI space?

d) Serial autocorrelation: The authors do not describe how they managed the risk of serial autocorrelation, even though they use small voxel clusters and a cluster–based method. This is essential to avoid false positives.

e) Other conventional steps are not described e.g. filtering.

3) Group ICA.

a) It is not acceptable to note that GIFT was used "with its pre–defined settings". GIFT is a large software platform that does not have "pre–defined settings". It has many choices available to the researcher in terms of algorithms, number of PCs, ICs. At a very granular level there are default settings, but this is after other choices have been made (e.g. one could use the default settings for the Infomax algorithm). The authors must describe in much more detail how they conducted the group ICA, especially given the small number of subjects.

b) Use of MDL to estimate the number of components seems unusual and unnecessary in this analysis. Did the authors perform a parameter sweep to think through their criteria for number of ICs given the study objectives?

c) Is a group ICA being performed or single subject? This is not clear – it appears single subject results are presented in figures.

d) ICA produces a substantial number of noise components such as the signals from CSF, vascular sources and pulsations, motion and so on. The authors do not describe how they removed the noise components and how many ICs remained (and what they were). This leaves the reviewer to wonder if the "cortex–wide" component they identify as corresponding to the calcium slow–wave signal is, in fact, one of these noise components. There are a number of figures where the activation maps resemble the output of ICA that corresponds to motion artifact.

e) The authors do not comment what ICs they have found – how many? What networks? Are the results expected for the rat brain? There are many excellent papers on brain networks derived from ICA in rodents they can compare results to.

[Editors’ note: following the rejection, the authors appealed. The appeal was assessed by the editors, and revisions were requested prior to acceptance.]

Thank you for submitting your article "Cortex–wide BOLD fMRI activity reflects locally–recorded slow oscillation–associated calcium waves" for consideration by *eLife*. Your article has been reviewed by four peer reviewers, one of whom is a member of our Board of Reviewing Editors and the evaluation has been overseen by Sabine Kastner as the Senior Editor. The following individual involved in review of your submission has agreed to reveal his identity: Sylvain Crochet (Reviewer #2).

The reviewers have discussed the reviews with one another and the Reviewing Editor has drafted this decision to help you prepare a revised submission.

Summary:

The manuscript by Schwalm et al., investigates a macroscopic signal equivalent of spontaneous slow wave events in the brain. This study has two major strengths: i.e. modeling calcium waves using a design borrowed from task fMRI (but applied to resting–state data), and an attempt to correlate a component identified using the data–driven technique of ICA with the "contrast" obtained from the former model, in the same data set. The approach of simultaneously recording calcium activity in the cortex and acquiring whole–brain functional MRI (fMRI) in–vivo is considered elegant, and the examination of the relationship between a defined neurophysiological event and resting–state fMRI (rsfMRI) signals is highly relevant and of great interest. As such, this work is timely.

However, the manuscript will require further analyses of the current data and all reviewers felt that the manuscript would benefit from scaling back on some of the claims and from discussing several missing references that are presently omitted. Specifically, it seems that some of the claims on the characteristics of the two brain states distract from the main strength of the paper – i.e. establishing a relationship between the calcium waves and the BOLD signal.

Essential revisions:

1) The authors postulate that resting–state fMRI (rsfMRI) networks are broadly synchronized networks upon the presence of global slow waves. To study the contributions of global slow waves to rsfMRI, the authors used isoflurane anesthesia (1.1–1.8%) to induce and maintain slow wave activity. However, the rsfMRI network in "slow–wave state" (Figure 4—figure supplement 4A) is not a typical network observed in previous studies, while the "active desynchronized state" induced by medetomidine is consistent with previously reported rodent rsfMRI studies. It would be important to show typical rsfMRI networks, such as interhemispheric/bilateral somatosensory, visual and auditory networks (1–3) for both experimental conditions. Using ICA analysis, which the authors have already done, to identify such typical rsfMRI networks will be sufficient to address this concern.

2) The isoflurane anesthesia used in this study was higher than previous rodent rsfMRI studies. In addition, Liu et al. showed that the interhemispheric somatosensory network gradually merged from a spatially specific network under light anesthesia (1.0% isoflurane) into a highly synchronized and spatially less specific network under deep anesthesia (1.8% isoflurane) (4). The reviewers appreciate the technical difficulty of this experiment and do not suggest additional experiments. However, given that the range of isoflurane levels as stated is very large, categorizing the individual animals based on their approximate experimental anesthesia levels would be very informative and should be added as a table, perhaps in the Supplement. Additionally, the authors should comment on how the different levels of anesthesia affected the network activity and how this was handled in interpreting their data.

3) The authors concluded that slow waves provide a temporal framework for long–range coherence. However, previous studies have also shown that long–range coherence exists under active desynchronized state (e.g., under medetomidine sedation). In fact, the authors identified default mode network (DMN) activity under this state. One could argue that DMN also demonstrates the characteristics of long–range coherence in the brain. We urge that the authors clarify in the text what they mean by long–range coherence (i.e., through slow wave) in the present manuscript, and they should also specifically discuss the difference(s) to other long–range networks that are also identified under active desynchronized state.

4) A recent paper by Ma et al. also examined the relationship between neuronal events detected through calcium recordings and resting–state hemodynamic signals through optical methods (5). They found clear evidence for the contribution of excitatory neural activity underlying resting–state hemodynamic patterns in the awake and anesthetized brain (i.e., interhemispheric/bilateral functional connectivity). Additionally, earlier work by Matsui et al., also utilizing simultaneous monitoring of neuronal calcium signals and resting–state hemodynamic signals identified two types of large–scale spontaneous patterns (i.e., global waves propagating across neocortex and transient co–activations such as interhemispheric connectivity) (6). Thus, the reviewer suggests two essential revisions. First, the present manuscript should acknowledge these prior works, discuss and compare the findings, as these prior studies are highly complementary. Furthermore, authors should also justify the use of the current analysis technique to examine the relationship between calcium signals and rsfMRI BOLD signals and they should also discuss why they used a different approach than the approach used by prior studies as the general objective were largely similar. Second, the comparison of BOLD signal extracted from 30 most active voxels and from an ROI (Figure 3 Suppl. 1) showed apparent delay in the peak and onset. Is this in anyway related to certain propagation characteristics as alluded by (6) and prior work done by the authors (7)? If so, the authors should characterize this phenomenon as it is highly interesting. In addition, a recent work by Leong et al. also demonstrated that the low frequency or slow waves can propagate robustly through long–range excitatory projections. Offering a multi–faceted view in the context of rsfMRI will definitely increase the impact of the present study.

5) At present, the authors attempt to claim that cortex–wide BOLD activity was identified only in slow wave state but not in the active desynchronized state and attribute their findings to different brain states by conducting the experiments under isoflurane anesthesia and medetomidine sedation. As eluded already above, this claim is generally too broad and simplistic. Moreover, the present finding is insufficient to support such assertions, unless awake animal experiments are conducted. In fact, this message distracts the reader from the core findings of the study, which is correlating a neuronal event (i.e. calcium slow wave activity) to brain–wide rsfMRI hemodynamic signals. The core message however, is not examining such relationship at different "states", which was the major reason the paper was rejected in the first place. The authors are certainly aware that brain states fluctuate temporally whereby different states are only dominant for a certain period. Moreover, certain features of the rsfMRI such as interhemispheric functional connectivity are also present at both states. Hence, it is a gross oversimplification to assign binary representations for the characterized brain states. Scaling back on these claims and better emphasizing the strength of the paper – i.e. relationship between the calcium waves and the fMRI will address the reviewers' concerns.

6) The authors stated that slow wave activity represents a brain state indispensable for memory consolidation. Knowing that the hippocampus plays a major role in memory consolidation, it would be interesting to also explore the BOLD fMRI signals in the hippocampus, but this is not an essential revision.

7) As stated in the previous review there was serious concerns related to the computational/analytic processing of the fMRI signal.

– The authors should provide a convincing argument that their "global signal" is not a processing or acquisition artifact. ICA decomposition always produces a substantial number of noise components from white matter, CSF, cardiac and respiratory sources. The authors do not mention if or how they dealt with them. They do not mention if their data was pre–whitened prior to submission to ICA. They do not present their component set and discuss it. Further, claims regarding the 'event–related' regressor and global signal would also have been burnished by thoughtful treatment of other potential extraneous signal noise such as drift, fMRI lag, temporal autocorrelation etc. Why has this global signal not been described in rsfMRI before, and if so, please cite the literature where such a "global signal" has been described before.

– It is difficult to understand why the rsfMRI ICA model suggests that under deep anesthesia, a neuronal component appears that represents wide activation. The rsfMRI ICA is not contingent on the calcium wave model – it was just regressed against it. Please explain this confusion.

– Please explain how your data unambiguously demonstrates that the fMRI signals come from neuronal sources.

– Please provide more methodological detail for the fMRI methods used.

– The pipeline used (HRF, task–based contrast modeling) is very vulnerable to artifacts. The link with neuronal activity is novel, and thus, it will be extremely important to be very cautious with making claims. This should be specifically addressed in the discussion.

– The authors should clarify whether they performed group or single subject ICA notwithstanding their presentation of single subject results.

– While the authors in their rebuttal provide some additional information as to their conduct of the ICA, this was not reassuring. At the heart of every ICA is the choice the researcher must make as to how many components to model. Model order is critical to the number and kind of components produced and the interpretation of results. Contrary to the authors assertion, the choice of model order is not "arbitrary". Please clarify were 4 or 17 or 22 PCs/ICs specified in the model? Were the number of PCs and ICs the same or different? Were different ICs used in different animals? If only 4 ICs were specified, one wonders if this drove the production of a single, ostensibly 'global' signal. If 17–22 ICs were specified, one wonders what all those other ICs were.

– The authors should discuss the significant drawbacks of ICA (as opposed to e.g. cluster or seed based methods) as it produces a large number of noise components along with actual gray matter maps. There is an entire field of research dedicated to the better, automated identification of components. The authors have not detailed any of their QC process, presented their component sets or even acknowledged the context and complexity of component identification in ICA. The authors need to discuss the issues associated with this approach.

---

## [Author Response]

[Editors’ note: the author responses to the first round of peer review follow.]

As you can see from the critiques the reviewers saw the technical advances of your study, but were concerned by the biological significance. It is indeed very important that you demonstrated that the waves extend globally, and that you can link the calcium imaging with the fMRI signal. However, the use of anesthesia is a major caveat. The use of anesthetics may explain the global spread, which might be different from natural SWS.

We want to point out that indeed isoflurane anesthesia was routinely employed in several high-impact studies to explore *aspects* of slow wave sleep. Most importantly, the rationale of this study was not to draw conclusions on natural SWS. It might very well be the case, that the global spread be different in SWS.

We rather asked, for the first time, whether we can use a neurophysiologically defined event – an up-state-associated slow wave – as a regressor for fMRI analysis. We only refer our results to the phenomenon of an individual Up state, and this Up state (represented as slow calcium wave) is clearly no epiphenomenon of anesthesia, as shown in multiple studies using a combination of single cell electrophysiology and calcium imaging (e.g. Chen, 2013).

Importantly, in a recent study, the group of Arthur Konnerth employed the same methodology as used in this paper, using the same anesthesia protocol, except for the use of cameras for recording calcium signals (Busche et al., 2015). They could indeed prove that the cortical synchronicity of slow waves is similar in natural sleep, compared to isoflurane. The treatment effective in restoring slow wave propagation in AD mice under isoflurane was effective in restoring SWS propagation (see Supplemental Figure 17 of the aforementioned publication). This again by no means should convey the notion that slow waves maintained under isoflurane are identical to SWS, but they seem to share important characteristics.

We see the general issue of studies bridging communities with apparently few interactions. In the fMRI community, oscillations of the BOLD signal have been attributed to a vast plethora of effectors: breathing, vascular effects, astrocytic activity, sub vs suprathreshold neuronal activity. Here, we provide for the first time a causal link between individual neuronal slow waves and an oscillation of the BOLD signal. Unlike mentioned by one of the reviewer, we do not merely correlate two signals, as performed in studies correlating EEG/electrophysiology and fMRI, which however are lacking the causal approach.

Certainly, we know, already since the pioneering studies of Steriade, that slow oscillations can be recorded from all cortical areas and certainly also from subcortical regions such as the thalamus.

However, it was far from clear, whether these events also dominate neurovascular signals. Therefore, we strongly believe, that this study bridges two communities which rarely interact, this notion was underscored by the high level of interest in this study both at imaging conferences (ISMRM) as well as at more basic science conferences (SfN).

We certainly agree that anesthesia can have tremendous effects on neuronal activity and on how wide-spread such activity might appear, e.g. upon sensory stimulation. We again want to clearly state that with our study we do not want to make statements regarding SWS or claim that anesthesia is similar to natural SWS and *we modified the manuscript accordingly*. Nevertheless, isoflurane anesthesia in the concentrations we administered has classically been used in many studies to mimic a certain type of neuronal activity which shares characteristics of activity also present under deep SWS (for a review see e.g. Destexhe et al., 2007). Many of the most cited studies in the field of slow waves are conducted under anesthesia: under isoflurane (Kerr et al., 2005; Luczak et al., 2007; Constantinople and Brunot, 2011; Grienberger et al., 2012.; Stroh et al., 2013; Chen et al., 2013;), but also under urethane (Steriade and Amizca, 1998) or Ketamin/Xylazin (Seamari et al., 2007). Here we do not aim to explain a phenomenon similar or mimicking SWS, but a specific brain state in which neurons of e.g. the cortex are undergoing phases of silence and phases of synchronous firing (also termed “up state” on the single neuron level, with our population calcium readout termed “slow wave”). These “on and off states” of neurons have been found to be meaningful for memory consolidation especially during SWS in behavioral (Ribeiro et al., 2004) and molecular (Yang et al., 2014) studies, but indeed also in studies conducted under isoflurane anesthesia it could be demonstrated that slow wave activity is meaningful for synaptic plasticity on the dendritic spine level (Chen et al., 2013). Despite the difficulties related with sleep studies in small rodent fMRI – where here even more, the specific sleep stage of deep SWS and not only the broader NREM classification as usually summarized in sleep studies, would have to be reached and verified – we here used isoflurane anesthesia to achieve clear and long enough on and off periods as described under the same anesthesia in previous studies (see above references) which is very difficult to achieve in natural sleep. In this way we are able to resolve a hemodynamic response upon an individual, separately detected slow wave, which is neurophysiologically defined, which to the best of our knowledge has not been done before. While we agree that the global spread might be exclusive to deep anesthesia (eventually generalizable to very deep SWS stages if so) we believe that this fact does not impede the biological value of our study as explained in detail further below.

To provide further evidence for the relevance of our findings, we now:

We directly contrasted neuronal responsiveness in slow wave state and another brain state – the desynchronized state – (see new Figure 2). We could conclusively show, that neuronal calcium responses differ fundamentally in these two brain states. Only in slow wave state, the entire cortex was recruited in slow oscillation-associated calcium wave activity, whereas in desynchronized state, the typical default mode networks could be identified using model-free ICA. This finding has immediate translation value, as it indicates, that resting state studies in humans, conducted under awake conditions, are very well captured in sedated desynchronized state in animals, but that in a slow wave state, the cortex is dominated by spontaneous slow wave events.

While we understand that the lack of thalamic involvement may be merely a technical issue, one reviewer felt that this could also be due to the anesthesia.

We published a study in Neuron in 2013, using the exact same methodology, that we can record slow waves from the thalamus (Stroh et al., 2013), but we conducted a new dataset:

In response to this comment, we now conducted thalamic recordings (see new Figure 5). Under our isoflurane anesthesia regime we could clearly show typical slow waves in thalamus, albeit with a lower SNR. And again, we could find a cortex-wide correlate to the thalamically recorded slow wave events, not surprising given the strong thalamocortical synchronicity (see Stroh et al., 2013). We again could not reliably detect BOLD activation in the thalamus, but we can now conclude that this will most likely represent a combination of technical issues: the surface coil not being sensitive in deeper brain regions, and presumably less neurons carrying the slow wave event, as indicated by the low SNR of the optical recordings. But most importantly, we can now exclude the major concern of the reviewer, the lack of thalamic BOLD activity being due to anesthesia as there is indeed thalamic slow wave activity under isoflurane.

We hope you can address these comments in a new manuscript that carefully discusses these caveats, emphasizes the technical advances and the importance of the biological significance of your finding.Also, please carefully explain the lack of thalamic activity, the rationale for using these particular anesthetics and the caveats associated with this.

We thank the reviewers for these valuable comments and concerns and now offer a new and fully revised version of the manuscript in which we address the mentioned critiques and rule out potential caveats pointed out by the reviewers, please see our point-by-point responses below.

Reviewer #1:

Default state functional MRI has become an increasingly important tool to explore various neurological and psychiatric conditions. This approach, which was pioneered by Dr. Marcus Raichle, can be used to characterize addiction, demonstrate developmental changes in cortical connectivity between different brain regions and identify developmental delays. The importance of this approach is documented by an increasing number of publications in high profile Journals. A major limitation of this approach is that the relationship between the BOLD signal and neuronal activity is complicated and there are those that believe this signal reflects primarily hemodynamic changes rather then actual changes in neuronal activities. One of the main technical limitations is that the use of MRI technology is difficult to combine with conventional electrophysiological techniques, because the magnetic fields are disturbed by the metallic nature of wires. The major strength of this study is the use of optical techniques that allows the combined characterization of neuronal signals and the wide-range Bold signal using fMRI approaches. The present study has also another strength in that it characterizes these activities in "small" animals. This has the major advantage that wide range optical signals can be characterized.

This study is clearly of great interest for a general readership: the demonstration that locally recorded calcium waves correlate with BOLD signals spanning distant cortical states is important since understanding the neuronal basis of cortical states is critical for understanding sensory processing, the generation of sleep and wake states and the role of memory consolidation, just to name a few examples. The study can relate slow wave onsets to fMRI fluctuations, which has not been possible before. These slow waves have been well characterized by other neurophysiological studies, which closes the loop between a well-characterized neuronal activity and the macroscopic fMRI signal. The study therefore strongly suggests that the cortex-wide BOLD signal is a property of slow-wave neuronal activity. The study seems to be well powered and carefully conducted. The introduction and discussion is very interesting to read and relates the finding in this rodent model to various general neurobiological phenomena. Thus, I don't have major concerns.

We thank the reviewer for his comments.

Reviewer #2:These authors combined fMRI with a fiber optic–mediated calcium recording method to study the slow calcium waves in the rodent brain. It is an excellent combination of techniques to study brain function at multiple scales. The main effort of this work is to decipher the potential linkage of slow calcium waves with the resting state fMRI signal of the anesthetized rodent brain, which is an important and novel direction for small animal fMRI studies. Besides the technical advancement, more discussion could be provided to better clarify the biological significance and image processing strategy. The reviewer has a few concerns needed to be clarified by the authors:1) The calcium signal was recorded in rats anesthetized with isoflurane. The trace of local field potential (Figure 1) mimics the burst suppression events. It will be better to discuss the states of the anesthetized animals of this work in comparison to the literature using similar drugs (or other anesthetics) to study slow wave activity.

We did now extend our discussion on these events; indeed we found very similar, if not identical, LFP signatures as others in the field (Kerr et al., 2005; Stroh et al., 2013; Busche et al., 2015).

2) The procedure to estimate hemodynamic response function (HRF) should be carefully described. One puzzling issue is the time course of the HRF, which showed very fast onset time in comparison to the existing models (Figure 2). The authors stated "Time course of averaged HRs exhibited a sharp rise 0.7 s {plus minus} 0.4 s after the onset of a slow calcium wave." What does the "sharp rise" mean (equal to onset time?). If it is the case, please discuss the early onset of the estimated HRF by referring to the existing literature on this matter. If the authors presented the time courses of simultaneously acquired calcium and fMRI signal, would they have observed any phase shift? It will be better to provide a cross correlation curve to specify the temporal features of the two kinds of signals.

The different temporal domains of the neuronal response and the neurovascular response represents the main enigma of fMRI, this is not different in this study. It is rather a strength of this study that despite this drastic difference in timing, we can extract a BOLD correlate to individual slow waves states.

A cross correlation between these signals seems not to be meaningful, due to these drastic differences in timing.

3) One of the key conclusions of this work is that the functional significance of the "slow calcium waves" specifies global correlation of the whole cortex. The authors suggested that the slow calcium waves were related to the slow wave activity as discussed by the authors. It will be better to explain the underlying rationale that no subcortical regions detected in the correlation maps.

The lack of subcortical BOLD activity is a mere technical issue due to the employment of a so-called surface MR coil, dedicated for high resolution of cortical areas, which have been the primary interest in our study so far.

We now conducted thalamic calcium recordings, see point 2.

Reviewer #3:In this manuscript Schwalm et al., have developed an innovative approach to study global patterns of activation brain–wide in relation to local slow–wave activity, combining local calcium imaging and fMRI in anesthetized rats. Using different analytical approaches, the authors show that cortical slow–waves detected locally with calcium imaging, are correlated to a global, cortex–wide activation detected with BOLD fMRI. This is an elegant and well-conducted study and I have little concern about the methodology – but I must say that I have little expertise in fMRI.My main issue with this study concerns the limitation of the interpretation of the results that need to be addressed in the discussion. Indeed, the authors make a strong statement about highly coherent slow–waves activity across the entire cortex based on their results obtained in deeply anesthetized rats. Anesthesia is not SWS. Although anesthetics have been used extensively to characterize slow–wave activity and determine its cellular mechanisms – because of the apparent similarity of the brain activity under some anesthetics and natural SWS – many studies also point directly or indirectly to critical differences between anesthetized states and natural SWS:

We do not want to give the impression and it was certainly not our intention to treat results obtained under anesthesia equally to SWS phenomena. We now rephrased the manuscript accordingly.

Nevertheless, as we wrote in the introduction and as the reviewer points out correctly, many studies addressing slow wave phenomena are conducted under anesthesia, particularly isoflurane. As mentioned above, in a recent study using the identical methodology, the group of Konnerth could show, that the cortical slow wave dynamics is similar in SWS and under isoflurane anesthesia. Again, this does not prove that they are identical, but that isoflurane anesthesia can be used to mimic certain aspect of slow waves occurring during deep sleep.

The novelty of our approach represents the approach to directly relate (not merely correlate) the two simultaneously measured signals, which is to use the neurophysiologically defined slow wave measured by the local calcium recordings as a predictor for the global hemodynamic BOLD signal in an event-related analysis.

We now strongly point out the fact that we do not want to compare our isoflurane anesthesia condition to SWS in the introduction and in the discussion and thus also moderate our conclusions as suggested by this reviewer, please also see new experiments 1-2.

1) Down– or Silent–states are shorter during SWS compared to anesthesia and last rarely longer than 1s (Steriade et al., 2001; Timofeev et al., 2001; Mahon et al., 2006; Chauvette et al., 2011). In this paper, the authors report patterns of activity with very long Silent–states (~ 5 s or more) between consecutive slow–waves, which indicates very deep levels of anesthesia. Although it is easy to understand the rational for maintaining such depressed cortical activity to provide enough temporal separation of single slow–wave events in this study, the relevance of the data acquire in such anesthetized state to natural SWS is questionable.

As in the multiple high impact studies using isoflurane to study up/Down states, we do not claim a one-to-one relation to SWS, and this was not the rational of this study. Indeed the reviewer is right; the long interval between the Up states is preferable for our analysis approach.

2) While under anesthesia slow–wave activity is highly synchronized across cortical areas, the long–range cortico–cortical coherence during natural SWS is much lower (Chauvette et al., 2011; Busche et al., 2015), probably due to the fact that slow–waves often occur very locally (Nir et al., 2011). As a result, many studies point to a decreased long–range functional connectivity during SWS compared to wakefulness (Olcese et al., 2016; Tagliazucchi et al., 2013). Therefore, the conclusion that slow–waves are highly coherent across cortex might only be true under anesthesia.

We are aware of these studies, indeed, the level of coherence might vary depending not only on anesthesia vs sleep conditions, but in addition might vary between different sleep stages. We never made the point that the level of coherence is similar under all these conditions; this was not the rationale of this study. We nevertheless want to mention that most of sleep studies refer to NREM sleep stages when talking about slow wave activity (Tagliazucchi et al., 2013 for humans and Olcese et al., 2016 for rats), which not necessarily resembles the pure stage of natural deep SWS as this is not easy to determine. In a modeling study, Deco et al., (2004) showed that if considering decrease of cholinergic neuromodulation to very low levels, slow waves do become global and resting-state networks merge into a single undifferentiated, broadly synchronized network, as it is the case in our study.

In future follow-up studies in naturally sleeping rodents, using the methodology implemented here, which is clearly not possible in the context of this study, there might be indeed different patterns of BOLD activation. But this is exactly the strength of our approach: we provide first evidence that individual slow waves show a clear BOLD correlate, so now this can be used to study this very question in the future.

3) SWS seems to promote, or even to be critical, for memory formation, whereas, to the best of my knowledge, anesthesia has a rather negative impact on memory formation.

We agree with the reviewer that important work regarding the role of slow waves for memory consolidation was conducted under natural SWS, on the behavioral (Ribeiro et al., 2004) as well as on the molecular level (Yang et al., 2014). We do not claim that in this state of anesthesia, memory formation takes place. However, we want to draw the reviewer’s attention on a study conducted by Chen et al., (2013) showing slow calcium waves present under isoflurane anesthesia to promote mechanisms necessary for synaptic plasticity on the dendritic spine level. In this study the authors show that the same synapses which are activated while mice are subjected to sensory stimulation are also active during spontaneous up states (the single-cell equivalent to our population slow waves) and identify recurrent NMDAR-activation-dependent calcium signals occurring reliably in a subset of spines in cortical neurons during slow waves under subsequent anesthesia in the same animals. These findings were conducted under isoflurane anesthesia and suggest that these calcium signals represent a cellular substrate for synaptic plasticity and therefore memory consolidation.

The authors should address these points in their discussion and moderate their conclusions.

We modified the manuscript accordingly.

Reviewer #4:What is the question asked by the manuscript? What is the answer provided? The manuscript appears as an elegant exercise to correlate slow brain activity with BOLD signal, I can't evaluate whether this is new. It doesn't seem new based on the references. There is nothing new in terms of biological data, the widespread and synchronized nature of slow oscillations is well known since the original description of the slow oscillation by Steriade et al., (1993). In those papers, also the synchrony with thalamus was established and then confirmed repeatedly, it is surprising that the thalamus shows no signal, slow oscillation in synchrony with cortex have also been shown in basal ganglia, why is there no signal in basal ganglia?

We now conducted a new set of experiments, please see experiment 3.

Why are these called calcium waves? Does it make reference to propagation? Or are these just increments in calcium signal with weak periodicity? Please clarify and be consistent throughout text.We call the recorded signals ‘calcium waves’ in accordance with others (Adelsberger et al., 2005; Stroh et al., 2013; Busche et al., 2016) indeed to reference to their propagation mechanisms as shown by others before (Stroh et al., 2013; Busche et al., 2016).I don't understand the statement in the subsection “Cortical slow wave–associated population activity can be reliably detected by optic calcium recordings during fMRI scans”, or what its significance is. ("This local confinement allows for a clear attribution of slow wave activity to a spatially restricted region").

With this statement we wanted to clarify calcium recordings derive a signal exclusively from the population of cells stained with the calcium indicator, when illuminated by the optic fiber. Considering this we can be sure about the source of our signal regarding local constrictions and can exclude influences from other regions, which is important as we wanted to relate a local slow wave event with a whole-brain readout and therefore need to be precise regarding the locality of such signals.

Also, the statement about the LFP is incorrect, if the LFP is recorded in bipolar configuration it is as confined or more than the calcium signal. Only in monopolar configuration (with a reference elsewhere in muscle or bone) the statement of volume conduction is true. In the original description of the slow oscillations by Steriade et al., (1993), the recordings are bipolar between the surface and depth of a cortical column and the intracellular recordings were obtained from the immediate vicinity of the LFP. By doing cross correlation between recording sites from spatially confined regions at different distances is how global synchrony was demonstrated, such demonstration would have been impossible with monopolar LFPs which are contaminated by volume conduction.Furthermore, the paper by Katzner et al., 2009 [cited?] reaches the opposite conclusion than Kajikawa and Schroeder (Cited in the manuscript) and states that the LFPs represent mostly local activity within 450 microns. The paper by Linden 2011, reaches yet another conclusion: that the area of tissue represented by the LFP depends on the degree of correlation of the underlying networks. Thus, the authors do a poor job here and this part of the manuscript should be rewritten.

We rephrased accordingly, but we clearly state that in principle LFP would also be a suitable method; we even show data on the one to one correlation. We use the optical method due to the fact that there is no interference between the fast shifting magnetic field of the MR scanner and the optic fiber-based readout.

The HR peak at 6.5 s contains at least 5 calcium waves, how is the HR related to an individual calcium wave? Calcium waves occur rhythmically and continuously, how is the effect of subsequent calcium waves eliminated from HR? What does the decay signal of HR represent? How can the HR signal decay many seconds after the calcium wave? Please explain.The most fundamental principle in the most accepted model of the BOLD fMRI signal (i.e. the HR) generation is that the BOLD signal is described by the convolution of the stimulus shape with the hemodynamic response function (HRF). The HRF, being the response to a delta function-shaped stimulus, has the major features of a signal rise to a maximum and a slower decay to baseline (accompanied by a small undershoot), and has a duration of several seconds. For longer stimuli, as a calcium wave in our case, the response is prolonged accordingly. In the commonly accepted General Linear Model, repeated stimuli (calcium waves in our case) add up and further prolong the HR. Our observations are therefore in agreement with the theory of BOLD fMRI that has been developed and tested over more than twenty years.The HR suggests that the entire brain is having increases and decreases of oxy/deoxy Hb, which are more than an order of magnitude longer than the underlying oscillations. How is this possible? What is driving the HR?

As described above, a HR lasting for several seconds is exactly what is expected from the theory of BOLD fMRI. The stimuli that drive the HR, are in our case the calcium waves. The cellular mechanism behind the HR is still a matter of debate in the field and its clarification way beyond the scope of this paper.

The finding that BOLD signal in the entire cortex is not new and confirms the early findings of Steriade et al., 1993. In the subsection “Calcium slow wave event–related analysis reveals cor tex–wide BOLD correlate”, the authors indicate that the data "suggest" but in reality the data "agrees" with previous descriptions.

We agree with the reviewer that our data agree with previous findings, in regards to the global spread of cortical waves. However, we for the first time showed that the BOLD correlate of cortical waves is linked to the calcium signal, and thus directly to the oscillatory activity. Indeed, Steriades seminal work showed slow waves (similar in SWS and under different types of anesthesia) to be present in distinct areas of the cortex. Nevertheless, a whole-brain correlate, showing how the activity of the entire cortex at the same time is directly related to an individual slow wave event, was never demonstrated before. What was indeed shown by others are local readouts in various brain areas or cortical areas at the same time (Steriade’s work, but also see the work of Igor Timofeev or Stroh et al., 2013) or the simultaneous presence of a slow EEG/LFP rhythm and the corresponding low frequency BOLD signals, which indeed show a correlation (Chang et al., 2013; Hsu et al., 2016; Magri et al., 2012 and many others), but the direct relationship of the whole-brain fMRI signal to an individual slow wave event was never analyzed before in the way we present it in this study.

Subsection “Calcium slow wave event–related analysis reveals cortex–wide BOLD correlate”: functional assemblies under isoflurane anesthesia? That sounds absurd. Widespread slow activity is the exact opposite of functional assemblies. The brain is anesthetized. Unless the authors refer to a functional state related with deep sleep? What would it be?

This statement was directed at a potential relation of isoflurane-associated slow waves to deep SWS oscillations. “On and off states” of neurons have been found to be meaningful for memory consolidation especially during SWS in behavioral (Ribeiro et al., 2004) and molecular (Yang et al., 2014) studies, but indeed also in studies conducted under isoflurane anesthesia it could be demonstrated that slow wave activity is meaningful for synaptic plasticity on the dendritic spine level (Chen et al., 2013). At least in SWS this reverberating activity has been associated with spike sequence replay related to previous experience (Ribeiro et al., 2004). Again, we only want to convey that in anesthesia, *aspects* of SWS can be studied.

This manuscript should not be published or considered further without clear answers and clarifications to the above questions.

We hope that our revised manuscript now gives sufficiently clear answers to the reviewers’ questions.

[Editors’ note: the author responses to the second round of peer review follow.]

Reviewer #1:The authors have done a great job in revising the manuscript. They added the characterisation of a desynchronised state, and they showed slow oscillations in the thalamus, which was a major concern in the previous submission. The authors use light anaesthesia, which was a confounding issue in the first submission, but the first submission did not show the desynchronised state, which differs fundamentally from the more synchronised state that has similar characteristics of SWS.The comparison with the BOLD signal is interesting and will contribute to our understanding of what neuronal activity actually underlies this signal.

We thank Reviewer 1 for his comments.

Reviewer #2:In this manuscript Schwalm et al., use an innovative approach to study global patterns of activation brain–wide in relation to "local" slow–wave activity, combining local calcium imaging and fMRI in anesthetized rats. Using different analytical approaches, the authors show that cortical slow–waves detected locally with calcium imaging, are correlated to a global, cortex–wide activation detected with BOLD fMRI.In this revised version, the authors also compare the pattern of activation during slow–wave activity in deep anesthesia to a more desynchronized brain activity under sedation. They find that the cortex wide activation during slow–waves contrast with the more restricted activation during sedation similar to the default mode network. The authors also studied the BOLD correlate of thalamic slow–waves, finding again a cortex–wide activation but no thalamic activation. The authors argue that the absence of thalamic activation during slow–waves could result from technical limitation.Although this revised version provide new and interesting data, the authors have not address my main concerns and I keep thinking that the relevance of this study is considerably limited by the use of deep level of anesthesia and technical limitation – in particular the very poor temporal resolution of fMRI. Hence, my comments for this revised version are very similar to my initial comments.However, I suppose this study might be of interest for the field of fMRI.

We thank the reviewer for his comments.

We agree on the reviewer’s concern regarding the low temporal resolution of fMRI, which is a general and unavoidable inherent drawback of this method. Nevertheless, it has to be stated, that the other reviewer suggested even lower temporal resolutions for our study, claiming this to be more appropriate for studies investigating resting brain networks. This by no means would be applicable for our experiments exactly because of the aim to resolve calcium slow waves during this type of ongoing activity, but clearly demonstrates the methodological, terminological and conceptional gap still existing between the communities.

As many rodent studies in fMRI are carried out under anesthesia out of practical reasons, the impact of different brain states induced by anesthesia (e.g. isoflurane slow waves versus desynchronized medetomidine sedation) on ongoing activity and stimulus response properties is of specific interest for this field. Also, here it is particularly relevant to consider readouts of local population activity as the calcium measurements and relate those signals to the broad fMRI signal. Directly relating a neurophysiological readout to fMRI BOLD is still not routinely done, even in the field of rodent fMRI. Especially because of the concerns the reviewers raise regarding globality versus locality of cortical engagement in slow wave activity it is of high relevance to pay attention to these local readouts. A local event, e.g. a change of excitability state of a small population of neurons, can have tremendous effects on global brain state and network responsiveness upon sensory stimulation. In the small animal fMRI community many studies of potential translational value are carried out in rodents, now more than ever due to the availability of genetic disease models and the possibility of optogenetic circuit probing. Especially resting state studies are performed under a plethora of different anesthesia protocols without controlling for the neurophysiological effects this might have on local cortical network activity and with referring to ongoing neuronal activity simply as ‘resting state fluctuations’ without further refinement of their properties. From this perspective it is of high value for the fMRI community to relate local network states induced by different types of anesthesia to the global BOLD response. But the value of our study goes beyond the specific relevance for fMRI measurements. To bridge the gap between the micro- and macro perspective on cortical activity held by the fMRI and the neurophysiology community, studies combining methodology and concepts of the two perspectives are relevant, also for a broader neuroscientific audience. As rodent so-called resting state studies carried out under undefined brain states induced by different anesthetics are compared and claimed to have translational value for human research, the induced states have to be closely investigated regarding their BOLD correlates. This will lead to a better understanding of which types of different states resting state or default mode activity consists, as this is also very loosely defined for the human brain. Tagliazucchi and Laufs, (2014) for example showed reliable drifts between wakefulness and sleep in typical fMRI resting-state data in humans, meaning that also in humans the exact state of the network during resting state fMRI is unknown and variable. Taking into account recent findings in rodents showing that also during awake resting a type of slow wave activity (which is also very likely different from sleep regarding length of silence periods and other properties) is present in different cortical areas (Fernandez et al., 2016; McGinley et al., 2015), the presence of slow wave activity and its effects on BOLD becomes even more relevant for resting state measurements and their potential translational value. Additionally, our study is not merely correlating two signals as many other studies did before, it is directly relating a neurophysiological event – the calcium slow wave – to its potentially present BOLD correlate by employing a specific type of fMRI analysis (eventrelated), which has never been shown before in such way. Thus, while we agree on all points the reviewer is rising in his comments and also can address them (see below), we truly believe in the relevance this study potentially has regarding cortical states and their different correlates, which is going well beyond the field of fMRI research. Such translational studies are valuable to reunify different fields investigating similar phenomena from their inherent point of view and eventually leading to a better defined terminology for said phenomena as well as how different observational scales of cortical activity can contribute to a holistic understanding of how the brain works.

My main issue with this study concerns the limitation of the interpretation of the results that need to be addressed in the discussion. Indeed, the authors make a strong statement about highly coherent slow–waves activity across the entire cortex based on their results obtained in deeply anesthetized rats and with poor temporal resolution. Anesthesia is not SWS. Although anesthetics have been used extensively to characterize slow–wave activity and determine its cellular mechanisms – because of the apparent similarity of the brain activity under some anesthetics and natural SWS – many studies also point directly or indirectly to critical differences between anesthetized states and natural SWS:

We completely agree with the reviewer and by no means wanted to give the impression to be in the belief anesthesia is comparable to slow wave sleep.

We also completely agree that with the temporal resolution at hand, a locally initiated, potentially traveling wave could not be resolved. We are aware of that such traveling waves exist, also e.g. under anesthesia (Stroh et al., 2013) and that the long-term aim of fMRI-based slow wave studies needs to be reaching a temporal and spatial resolution which enables a brain-wide readout of such traveling activity. This up to now is not possible with the current methods, which is why we aimed at resolving a cortical BOLD correlate of calcium slow waves by employing a specific type of analysis. We are relating every calcium event detected in the optical signal to the brain wide BOLD activation of the animal and thereby detect the reported cortex-wide activation. This analysis reveals the ongoing BOLD signal in the whole cortex related to a locally detected wave in the calcium signal. It is not revealing a correlation, nor is it similar to coherence or connectivity and therefore cannot be directly compared to results obtained by these measures. We agree that for clarifying these issues we need to modify the discussion and especially focus on the limitation of the interpretation of the results.

1) Down– or Silent–states are shorter during SWS compared to anesthesia and last rarely longer than 1s (Steriade et al., 2001; Timofeev et al., 2001; Mahon et al., 2006; Chauvette et al., 2011). In this paper, the authors report patterns of activity with very long Silent–states (~ 5 s or more) between consecutive slow–waves, which indicates very deep levels of anesthesia. Although it is easy to understand the rational for maintaining such depressed cortical activity to provide enough temporal separation of single slow–wave events in this study, the relevance of the data acquire in such anesthetized state to natural SWS is questionable.

It is indeed true that during SWS, down or silent states are much shorter than during deep isoflurane anesthesia. This faster reoccurrence of the waves may also be the reason for a more local than global spreading of this activity. In the manuscript indeed we focused on data with higher temporal separation to detect single slow-wave events as the reviewer correctly pointed out. Again, we do not want to draw a parallel from our deep anesthesia data to natural slow wave sleep and apologize to have given that impression throughout the manuscript. Our only interest in this study was to analyze the whole-brain correlate of such locally occurring slow wave events. The interrupting silent states may be of different length in different types of slow wave rhythms, but the slow wave event itself by its very own nature and microarchitecture is comparable throughout states on a neurophysiological level as– no matter if in natural slow wave sleep, anesthesia or during awake resting- on the cellular level it consists of synchronous firing of a population of thalamic and cortical neurons.

2) While under anesthesia slow–wave activity is highly synchronized across cortical areas, the long–range cortico–cortical coherence during natural SWS is much lower (Chauvette et al., 2011; Busche et al., 2015, Fernandez et al., 2016), probably due to the fact that slow–waves often occur very locally (Nir et al., 2011). As a result, many studies point to a decreased long–range functional connectivity during SWS compared to wakefulness (Olcese et al., 2016; Tagliazucchi et al., 2013). Again, many studies point to slow–wave events being rather local than global during natural slow–wave sleep. Therefore, the conclusion a cortex–wide activation during slow–waves events might only be true under anesthesia.

We are well aware of the existence of such locally occurring slow waves in lighter anesthesia states or even when the animal is resting but awake, from the literature, and this is also what we are seeing in data from our own lab. Nevertheless, what we are looking at is different from measures of functional connectivity and we would of course agree and expect that compared to wakefulness, functional connectivity in SWS as well as during deep anesthesia is much lower. We definitely agree that we have to address the issue of rather local slow waves during natural slow wave sleep and that therefore our results may be valid for deep anesthesia states only, in the discussion. Also we want to further discuss the point that what we are observing might be several, locally initiated, but traveling wave.

3) SWS seems to promote, or even to be critical, for memory formation, whereas, to the best of my knowledge, anesthesia has a rather negative impact on memory formation. In order to synchronize cortical activities across areas and efficiently drive plasticity, the precise timing – at milliseconds scale – of slow–waves is critical (Miyamoto et al., 2016). Such time precision cannot be achieved with fMRI.

We agree about the millisecond scale timing necessary to synchronize cortical activities across areas and efficiently drive plasticity as shown by Miyamoto et al., 2016. Indeed, such a temporal precision cannot be reached with fMRI. Regarding the meaningfulness of slow wave events under anesthesia for memory formation we do not want to make any claim in our manuscript.

Nevertheless, we want to draw the reviewers attention on a paper from Chen and colleagues (2013) which showed that dendritic spines in neurons are active during the neuron’s up state (and inactive during silent states) and that spontaneously recurring up states evoked in these spines under isoflurane anesthesia “patterned” calcium activity that may control consolidation of synaptic strength following epochs of sensory stimulation. So it may be the case that at least the up state on the single neuron level under isoflurane anesthesia indeed plays a role in the consolidation of synaptic strength.

The authors should really address these points in their discussion and moderate their conclusions.

We fully agree with the reviewer on this point and prepared a revised manuscript addressing these points in the discussion and moderating the conclusions drawn from our data.

Reviewer #3:Essentially, it seems to me the authors are trying to prove that there is cortex–wide BOLD activation in the rat brain that is driven by (or at least correlates to) activation in calcium slow–wave response by means of demonstrating that one can convolve calcium slow wave tracings with a FIR basis set and fit this to the BOLD data. However, FIR basis sets are complex alternatives to standard models of the HRF and while they allow a large amount of flexibility this comes at the price of bias–variance tradeoff, and they should be used within rigorous multivariate models. Generally, FIR basis sets are used in studies aiming at characterizing the shape of the HRF, and not for studies aiming to detect activations. This study aims at detecting activation. As well, there are a very small number of subjects, and many substantial questions in the methods used to process the BOLD, ranging from sequence used to pre–processing to use of ICA. It is not clear to me what the authors have actually "detected" in the BOLD – it may just be motion, or other artifact.

We thank the reviewer for the comments on our manuscript. Nevertheless, we have to state that many assumptions the reviewer makes are factually wrong. Regarding the main criticism expressed here that “Generally, FIR basis sets are used in studies aiming at characterizing the shape of the HRF, and not for studies aiming to detect activations” we totally agree with this view. What the reviewer does not take into consideration is that we did exactly what she/he claims what should be done with this basis function set. We used the FIR basis set to derive the hemodynamic response upon the onsets of optically detected slow calcium waves. Afterwards, we use the hemodynamic response that was detected to convolve it with a binarized slow calcium wave vector that is used afterwards as a regressor. This second step of course was performed in a different subset of animals to avoid what is known as “double dipping” (this is explicitly informed in lines 620-635 of the main manuscript). Furthermore, the sentence is misleading, since an FIR basis set is used to characterize the HR, not the HRF, since an HRF is already a basis function itself. We therefore assume the reviewer did not understand the important difference between HR and HRF, but also we have to say that nevertheless we already did exactly what the reviewer claims what has to be done: characterizing the shape of the hemodynamic response upon a slow calcium wave. Therefore, we conclude that the reviewer did not understand the procedure used by the authors to perform the GLM, and consequently further claims on this issue are obsolete. Additionally, the claim that our results could be due to a motion artifact is not a reasonable assumption considering that the data was acquired in deeply anesthetized animals which tend to move very little due to the deep anesthesia. Nevertheless, for the GLM data we carried out a series of control analysis to reveal eventual artifacts leading to spurious activation, as e.g. switching around the regressor obtained from the calcium data temporally or switching calcium regressors between animals – both without any results, showing that the activation we detected based on this regressors are very specifically related with the underlying neuropyshiological event (see in detail Figure 4—figure supplement 2). And, additionally for the results obtained with the ICA method we showed that the ICA component does strongly correlate with the regressor of the GLM model used for the same animal. This strong correlation is significant for all the animals, which totally annihilates the idea of a spurious component.

Specific Methods ConcernsI have substantial concerns with the methods described to acquire, pre–process and analyze the resting–state fMRI data. These include but are not limited to:1) Sequence: a very short TR of 1 sec was used. This is unconventional and has disadvantages in the acquisition of resting–state (in rodents, or humans). Generally, a TR of at 2–3 secs is preferred.

To our knowledge, most of resting state fMRI experiments in rodents use a TR between 0.5 and 2 seconds which is also stated in a recent review by PAN W-J and collaborators (2015): “As in human rs-fMRI studies, most rodent studies employ fast imaging sequences to image the brain on the order of once per second… A typical rs-fMRI study in rats has an in-plane spatial resolution of 200–400 microns, a slice thickness of 1–2 mm, a repetition time (TR) of 0.5–2 s”.

2) Pre–processing pipeline: this is (thinly) described as being realignment, reslicing and smoothing. This raises the following concerns:a) Coregistration and normalization do not appear to have been performed prior to smoothing. Realignment only performs within–subject motion correction. It does not perform between–subject motion correction. The authors describe a process (subsection “fMRI analysis”) where they take heavily processed maps and manually align them with anatomical images (presumably for purposes resembling coregistration) using ImageJ. This is of concern because (a) coregistration should be performed earlier in the pipeline, prior to smoothing (b) They are performing a manual process and not using an appropriate normalization algorithm and (c) Using a piece of software intended for the processing of visual (not neuroimaging) images. The authors should re–perform their pre–processing pipeline to incorporate coregistration and normalization prior to smoothing. Excellent choices are readily available in SPM12, which they are using.

The reviewer claims that the authors presumably co-registered images using Image-J. The reviewer expresses this concern as this would not correspond to a proper normalization. Here, the reviewer is mistaken since the authors did not perform any alignment. We clearly mentioned this in the subsection “fMRI analysis “cited by the reviewer that 20 beta-maps were overlaid, so a co-registration was not necessary in this case. Again, in same subsection, the reviewer mistakenly argues that the entire pre-processing pipeline should be re-performed which is based on her/his misunderstanding of the data analysis. Again, we want to strongly state, that since no group analysis was performed, neither co-registration nor normalization is necessary or reasonable for our preprocessing pipeline.

Because no group analysis was done between-subject motion correction is not necessary since no second level analysis was ever performed. Statistical tests had only been performed in within subject analysis, so no co-registration, neither normalization of datasets is necessary.

b) Smoothing has the effect of boosting SNR. It is generally used in pipelines where masses of voxels are being considered in larger numbers of subjects. Here, a small number of voxels is being considered in a small number of subjects. How do we know the resulting 'signal' is not artificially boosted by the smoothing process?

The reviewer correctly argues that smoothing can sometimes artificially increase the SNR. Nevertheless, what the reviewer does not consider is the size of the smoothing applied. Typically, a large smooth can boost up the SNR, nevertheless the reviewer does not consider that the voxel size in our case is 350 x 325μm² spatial resolution (slice thickness 1.2mm) and the smoothing used is 500μm, what practically cannot increase the SNR by any means.

c) The lack of normalization and use of manual methods in ImageJ introduces many concerns regarding anatomic standardization among the various parts of this complex pipeline. It is not clear to this reviewer how integrity is maintained in comparing one location vs another between processes and animals. For example, the sites of optical fiber recordings, ROI, etc. – why have conventional methods not been used to place all locations in MNI space?

The reviewer argues that it is not clear how the integrity in the sites of optical recordings was maintained, while it is clearly stated in the manuscript how and where the craniotomies were performed (S1, subsection “Dye or virus injections and fiber placement”), which is obviously consistent throughout the entire study.

The reviewer suggests putting all the data in MNI space. This suggestion again highlights the fact that the reviewer did not understand the experimental protocol. Standardization of brains in MNI space is only useful for group analysis but is not necessary and furthermore, definitely not recommended for single subject analysis as those performed in our manuscript, since they unnecessarily distort the brain volume to make it fit onto a template.

d) Serial autocorrelation: The authors do not describe how they managed the risk of serial autocorrelation, even though they use small voxel clusters and a cluster–based method. This is essential to avoid false positives.

Serial autocorrelations were performed by using an autoregressive AR(1) model during classical (ReML) parameter estimation. These are standard SPM12 parameters.

e) Other conventional steps are not described e.g. filtering.

All datasets had been filtered using a Highpass Filter with a cutoff frequency of 1/128 seconds. As well these are the standard SPM12 parameters.

3) Group ICA

This point of the reviewer is entitled “Group ICA”, which raises the authors concern that this reviewer may have misunderstood the point of the manuscript. We would like to emphasize that we did not perform any kind of group analysis in this study and do not understand from where she/he got that impression.

We did this type of analysis (single-subject ICA) specifically to avoid the mentioned danger of spurious results due to a reduced number of subjects or due to the co-registration and normalization procedures. On the contrary, we show that our BOLD activations and independent components in the ICA analysis are robust and reproducible on a subject-by-subject basis for all the animals.

a) It is not acceptable to note that GIFT was used "with its pre–defined settings". GIFT is a large software platform that does not have "pre–defined settings". It has many choices available to the researcher in terms of algorithms, number of PCs, ICs. At a very granular level there are default settings, but this is after other choices have been made (e.g. one could use the default settings for the Infomax algorithm). The authors must describe in much more detail how they conducted the group ICA, especially given the small number of subjects.

We indeed do not mention the details of the ICA parameters used, but those a rarely mentioned within a manuscript. Nevertheless, we can add them here: on preprocessing, means were removed per timepoint. The default mask and standard PCA (with 2 data reduction steps) have been used. For back reconstruction, GICA has been used and all results have been scaled by their Z-scores. In the first step, 4 PC/IC has been used, none in the second. We used the Infomax-algorithm with regular stability analysis and no autofill was performed. The options for the Infomax-algorithm were as follows: Block:119, Stop:1e-006, Weight:0, Lrate:0.01082, MaxSteps:512, Anneal:0.9, Annealdeg:60, Momentum:0, Extended:0, Posact:off, Spering:on, Bias:on Verbose:on. The reviewer asks for a detailed explanation of the performed group ICA which we never performed and which would make no sense to perform it in the framework of our study.

b) Use of MDL to estimate the number of components seems unusual and unnecessary in this analysis. Did the authors perform a parameter sweep to think through their criteria for number of ICs given the study objectives?

Regarding this point, we indeed used MDL to estimate the number of components because it is an option that is part of the GIFT platform, which is commonly employed. We agree with the reviewer that it is not “necessary” for this analysis but it is not detrimental for it either. Indeed, most of other authors usually choose an arbitrary number of components – 15-20 or even 40 – for the ICA and do not use the MDL procedure. In our case, while using MDL most of the animals showed between 17-22 components what is practically similar to choosing them arbitrarily.

c) Is a group ICA being performed or single subject? This is not clear – it appears single subject results are presented in figures.

After exposing the criticisms in the three previous points concerning mistakes in the supposedly performed “group ICA” analysis of our data, only now the reviewer wonders if any group ICA was ever performed. Just to be clear on this point: no group ICA was performed in this manuscript at all. Actually, the only time the words “group ICA” appear together in the manuscript is when the tool GIFT is cited, which can indeed also be used for such analysis, but does not have to be. In any case as stated throughout the manuscript, we used it for single-subject ICA. Finally, and just to complement, as the reviewer is speculating on this point, all the figures are single subject ICA analysis because we only performed single subject ICA analysis.

d) ICA produces a substantial number of noise components such as the signals from CSF, vascular sources and pulsations, motion and so on. The authors do not describe how they removed the noise components and how many ICs remained (and what they were). This leaves the reviewer to wonder if the "cortex–wide" component they identify as corresponding to the calcium slow–wave signal is, in fact, one of these noise components. There are a number of figures where the activation maps resemble the output of ICA that corresponds to motion artifact.

The reviewer argues that the authors do not show all the components and that therefore it would be difficult to know whether the “cortex-wide” component is indeed a true or just a noise component. In figure 4—figure supplement 4A-B, we show the signal of the “cortex-wide” ICA component to strongly correlate with the regressor of the GLM model used for the same animal. This correlation is significant for all the animals and this is clearly stated in the manuscript. The fact that this correlation is strong totally annihilates the idea of a spurious component. The authors cannot be sure about the reviewer having understood this point since the reviewer was never sure if a single subject ICA was performed or not.

e) The authors do not comment what ICs they have found – how many? What networks? Are the results expected for the rat brain? There are many excellent papers on brain networks derived from ICA in rodents they can compare results to.

Here, the reviewer wonders which components were found and if these results are expected in the rat brain under medetomidine. This point is explained in detail in figure 4—figure supplement 4C where the rat DMN can clearly be observed in two animals. In the text this issue is covered in detail in the subsection “Calcium slow wave event-related analysis reveals cortex-wide BOLD correlate” of the main manuscript, where we indeed compare these results to the mentioned papers on brain networks derived from ICA in rodent (yes these results are expected, yes they match the results of others).

[Editors' note: revisions were requested prior to acceptance, as described below.]

Summary:The manuscript by Schwalm et al. investigates a macroscopic signal equivalent of spontaneous slow wave events in the brain. This study has two major strengths: i.e. modeling calcium waves using a design borrowed from task fMRI (but applied to resting–state data), and an attempt to correlate a component identified using the data–driven technique of ICA with the "contrast" obtained from the former model, in the same data set. The approach of simultaneously recording calcium activity in the cortex and acquiring whole–brain functional MRI (fMRI) in–vivo is considered elegant, and the examination of the relationship between a defined neurophysiological event and resting–state fMRI (rsfMRI) signals is highly relevant and of great interest. As such, this work is timely.

We thank the reviewers for this summary of our work. Investigating the relationship of neurophysiologically defined calcium wave events and the fMRI BOLD signal was indeed the main aim of our study. We now brought this objective into the focus of the manuscript and further highlighted the approach of correlating the pan-cortical activation component detected by model-free ICA with the GLM regressor.

However, the manuscript will require further analyses of the current data and all reviewers felt that the manuscript would benefit from scaling back on some of the claims and from discussing several missing references that are presently omitted. Specifically, it seems that some of the claims on the characteristics of the two brain states distract from the main strength of the paper – i.e. establishing a relationship between the calcium waves and the BOLD signal.

We fully agree with the reviewers that the relation between slow wave events and BOLD fMRI represents the essence and novelty of this work. We therefore significantly scaled back on our claims in relation to brain states. However, we still feel the need to contrast the bimodal nature of slow waves with a network activity in which these large amplitude slow waves are absent, for demonstrating the specificity of our findings. We refrained from the nomenclature of “desynchronized state” as this might indeed be misleading. The term “desynchronized” came into being as a description of the calcium traces, in line with the electrophysiology literature (e.g. Poulet and Petersen, 2008; Constantinople and Bruno, 2011; Eggermann et al., 2014). However, this indeed may not fittingly describe the large-scale network dynamics observed in BOLD fMRI measures. We rather now refer to this activity as “persistent”, unimodal activity (Constantinople and Bruno, 2011), which still describes the nature of the local network behavior reflected by the calcium readout but is not misleading regarding the BOLD activity. Indeed, during “persistent activity”, we do find spatially separated, specific network signatures such as the default mode network (Hsu et al., 2016b; Lu et al., 2012), highlighted in the revised version of the manuscript. We refrain from using the term “state”, as there might be different sub-states of both types of activity, and clearly, to describe and characterize brain states was not the main scope of this paper. We modified the figure set accordingly, and focus solely on slow wave activity in the main figures. Nonetheless, we feel that these experiments in the conditions of persistent, unimodal network activity are highly useful not the least as a control, strengthening the specific relation of slow wave events and their related BOLD fMRI correlate.

In response to the reviewer´s comments, we conducted several new analyses of the current data, focusing on the relationship of BOLD and slow calcium waves, and providing additional evidence for a specific relation of slow calcium wave events and fMRI BOLD by ruling out spurious sources, particularly in our ICA approach:

1) As mentioned above, the ICA component showing pan-cortical activation (new Figure 5) in slow wave state is highly correlated to the GLM timecourse derived from optically detected calcium waves (new Figure 5). Nevertheless, we share the reviewers concern regarding the ICA approach and now conducted group ICA for the slow wave datasets and now show the complete component set (new Figure 5—figure supplement 1). We identified components showing pan-cortical activation, both on the individual animal level (new Figure 5—figure supplement 1), and in the group analysis (new Figure 5—figure supplement 1). Additionally, we show that exclusively this pan-cortical component is related to higher power in the ultra-low frequency range (0-0.1 Hz) as expected for slow wave activity (new Figure 5—figure supplement 2).

2) We already demonstrated that temporally mirroring the slow wave vectors of each dataset for obtaining the GLM regressors leads to a complete absence of BOLD, suggesting a non-spurious source of the observed pan-cortical activation. Using temporally unmatched regressors (Figure 3—figure supplement 3) does not only lead to a complete lack of cortical activation (Figure 3—figure supplement 3), but also these GLM regressors do not correlate anymore with the slow-wave related ICA component when using the cross-correlation approach (new Figure 5). As we are well aware of the importance of these new cross-correlation controls, we now show them in main Figure 5.

Additionally, we prepared an analysis showing that if the timecourse derived from the slow wave vector is regressed out of the data, the subsequent ICA does not show any pan-cortical components anymore (new Figure 5—figure supplement 2).

3) We conducted seed-based analysis placing a seed ROI in the somatosensory cortex, the location we recorded the neuronal calcium transients from. Notably, confirming the slow calcium wave event-related analysis, also with the seed-based approach we again find pan-cortical activation during ongoing slow wave activity (new Figure 6). The mean signal intensity in the seed-ROI also correlates high with the normalized signal of the previously identified pan-cortical component (new Figure 6; new Figure 6—source data 1 – Table 2). Ongoing persistent activity on the other hand is characterized by spatially separated foci of BOLD activity showing typical connectivity patterns reflecting the inter-hemispheric connectivity between these sensory areas (e.g. Hodkinson et al., 2017) as shown in new Figure 6—figure supplement 1 and new Figure 6—source data 1 – Table 2. Finally, we now highlight more an optical calcium recordings-informed correlation analysis (new Figure 6): when hemodynamic responses (HRs) upon previously detected slow calcium wave onsets in a ROI in somatosensory cortex are correlated with every remaining voxel in the brain, again a cortex-wide network becomes apparent. Independent of the location of the somatosensory ROI (right or left hemisphere) this correlation analysis revealed highest r-values for the entire cortex (Figure 6; Figure 6—figure supplement 3).

4) We extended the analysis of network activation patterns. We now show that, independent of the condition, typical rs-networks can observed (new Figure 5—figure supplement 1 and Figure 5—figure supplement 4), as described previously (Hsu et al., 2016b; Lu et al., 2012, Jonckers et al., 2011).

5) We conducted seed-based analysis to investigate potential hippocampal recruitment during slow wave activity, which indeed seems likely (Hahn et al., 2006; Chan et al., 2017). Employing an atlas-based selection of the seed-ROI encompassing the bilateral hippocampal formations, we now correlated activity in the hippocampal ROI to cortical BOLD (new Figure 6—figure supplement 2). This analysis revealed a pan-cortical correlation of the hippocampal ROI during slow wave activity.

During persistent activity, the pan-cortical BOLD correlation was absent, but frontal areas correlated with the hippocampus as it has been shown during awake or more active states (Sigurdsson et al., 2010; Li et al., 2015).

6) We added the missing references and discussed these studies appropriately.

Essential revisions:1) The authors postulate that resting–state fMRI (rsfMRI) networks are broadly synchronized networks upon the presence of global slow waves. To study the contributions of global slow waves to rsfMRI, the authors used isoflurane anesthesia (1.1–1.8%) to induce and maintain slow wave activity. However, the rsfMRI network in "slow–wave state" (Figure 4—figure supplement 4A) is not a typical network observed in previous studies, while the "active desynchronized state" induced by medetomidine is consistent with previously reported rodent rsfMRI studies.

Slow wave activity has long been described as a multiscale phenomenon, present on the single neuron level with the typical bimodality, rather hyperpolarized Down states and rather de-polarized and stereotypical Up states (e.g. Sanchez-Vives, Massimini and Mattia, 2017). These transitions between Up and Down states can occur synchronously in larger populations as identified by optic-fiber-based calcium recordings or LFP recordings (which represents the most common readout for this type of activity, starting from the seminal experiments of Mircea Steriade) and can even be detected in whole brain readouts as EEG (see for instance Massimini et al., 2004). In all the mentioned readouts if by patching a cell or looking at ensemble activity – and this may be in vivo or in vitro recordings – the most prominent feature of this activity is bistability: the reoccurrence of a stereotypical spontaneous neuronal response – slow waves – while between these responses the network remains silent. From previous work we know that these waves may travel across the cortex as well as to subcortical areas (e.g. Stroh et al., 2013; Massimini et al., 2004), a property which we cannot resolve with BOLD fMRI because of its low temporal resolution (1s of fMRI sampling rate versus 30-37mm/s wave propagation). Still, it is of immense interest for a broad neuroscientific community what the actual BOLD correlate of this activity might be and which brain areas are recruited by a locally recorded wave. It is indeed highly likely that the propagation of these waves relies on spreading activity of local networks (Sanchez-Vives, Massimini and Mattia, 2017), which is why we correlated the BOLD activity to a locally precise readout of neurophysiologically defined slow wave events based on calcium transients.

We furthermore agree with the reviewer, that indeed the direct BOLD correlate of the slow wave activity has not been identified in earlier studies, as this requires the very analysis we put forward in our manuscript. But nonetheless, a pan-cortical/cortex-wide activation was reported in previous work using similar anesthetic regimens (Kalthoff et al., 2013), and it has been speculated, that a pan-cortical activation could have its source in slow wave activity, but it could not unambiguously be assigned to this phenomenon (Liu et al., 2011; Kalthoff et al., 2013; Pan et al., 2013).

Congruently, during persistent activity, which supposedly reflects a network which is more active and therefore richer in complexity, lacking the bistability of slow wave activity, we find typical spatially complex networks such as the default mode network, referred to as “resting state networks”, as identified in many other studies.

It would be important to show typical rsfMRI networks, such as interhemispheric/bilateral somatosensory, visual and auditory networks (1–3) for both experimental conditions. Using ICA analysis, which the authors have already done, to identify such typical rsfMRI networks will be sufficient to address this concern.

We thank the reviewer for this suggestion. We analyzed our data accordingly, and as mentioned above, for both experimental conditions could indeed identify typical rsfMRI networks previously described (see new Figure 5—figure supplement 1 and Figure 5—figure supplement 4). In addition to the already identified default mode network activation (Hsu et al., 2016b; Lu et al., 2012, Jonckers et al., 2011) observed during persistent activity (new Figure 5—figure supplement 1 and Figure 5—figure supplement 4), we identified another component putatively recruiting resting state auditory networks involving sensorimotor (medial superior) and auditory (inferior arietal) cortices (new Figure 5—figure supplement 4).

During slow wave activity, additionally to the pan-cortical component, by using ICA we can now also identify networks resembling typical resting state patterns, as e.g. auditory and visual cortex, as well as hippocampus.

These results are in line with the work of others characterizing rat resting state networks (e.g. Jonckers et al., 2011)

2) The isoflurane anesthesia used in this study was higher than previous rodent rsfMRI studies. In addition, Liu et al., showed that the interhemispheric somatosensory network gradually merged from a spatially specific network under light anesthesia (1.0% isoflurane) into a highly synchronized and spatially less specific network under deep anesthesia (1.8% isoflurane) (4). The reviewers appreciate the technical difficulty of this experiment and do not suggest additional experiments. However, given that the range of isoflurane levels as stated is very large, categorizing the individual animals based on their approximate experimental anesthesia levels would be very informative and should be added as a table, perhaps in the Supplement. Additionally, the authors should comment on how the different levels of anesthesia affected the network activity and how this was handled in interpreting their data.

We used relatively high levels of isoflurane concentrations as we were aiming to achieve slow wave activity around 0.1 Hz. As depth of anesthesia is highly dependent on the physiological state of the animal we relied on the online calcium signal readout we had available to ensure stable slow wave activity throughout the measurements and adjusted the isoflurane level accordingly, to achieve a rather uniform occurrence of slow waves. Consequently, while the absolute level of isoflurane anesthesia varied, based on many parameters such as length of anesthesia, we achieved rather constant conditions in terms of the characteristics of slow waves.

We already showed in earlier work, that by varying anesthesia levels on intra-individual level, we indeed observed a direct relation of the level of anesthesia and the occurrence of slow waves (see Figure S4 panel E in Stroh et al., 2013. Neuron)

Furthermore, we feel that our results perfectly match the findings of Liu et al.: in lightly anesthetized or sedated condition – termed persistent activity based on our real-time calcium recordings – we also identified the typical rat default mode network. We, however, use the calcium recordings to stably maintain slow wave (or persistent) activity.

Please find below the table of the respective anesthesia levels for the reviewer´s appreciation. We would, however, not show this table in the manuscript, as this might mislead the readers. The strength of our study lays in our approach to maintain a very stable network activity using calcium recordings, not relying on variable isoflurane levels. As mentioned above, the slow wave occurrence as identified in the calcium recordings is the main variable, and the different anesthesia levels were adjusted to maintain these values. Therefore, while we present this very table here, we would not deem it useful to include it in the manuscript, but would of course follow the reviewer´s suggestion, if he/she still deems it important.

**Experiment****Mean SW frequency (Hz)****Isoflurane (%)**C1E10.171.6C1E20.161.3C20.171.1C30.221.6C3a0.161.8C40.181.5C50.11.7T10.181.5T20.21.4

3) The authors concluded that slow waves provide a temporal framework for long–range coherence. However, previous studies have also shown that long–range coherence exists under active desynchronized state (e.g., under medetomidine sedation). In fact, the authors identified default mode network (DMN) activity under this state. One could argue that DMN also demonstrates the characteristics of long–range coherence in the brain. We urge that the authors clarify in the text what they mean by long–range coherence (i.e., through slow wave) in the present manuscript, and they should also specifically discuss the difference(s) to other long–range networks that are also identified under active desynchronized state.

We apologize for this confusion and agree with the reviewers regarding the need for clarification on terminology and proposed function. As coherence is mainly an EEG-related term we decided to refrain from this terminology. The bistable “default” state of slow oscillations or slow waves may be of low complexity (Sanchez-Vives, Massimini and Mattia, 2017; Bettinardi et al., 2015), but cortically it can “spread smoothly like an oil-spot” (Sanchez-Vives, Massimini and Mattia, 2017), eventually recruiting the entire cortex.

A recent study by Leong et al., (2016) showed that low-frequency slow waves can propagate robustly through long-range excitatory projections e.g. from the thalamus to the cortex, which is not a contradiction to the aforementioned idea that slow wave activity can also spread cortico-cortically from cell assembly to cell assembly.

As mentioned by the reviewer, the persistent activity is mainly characterized by a spatially specific pattern of network activity (see new Figure 5—figure supplement 4), likely relying on long range connectivity. We added up upon this topic in the Discussion.

4) A recent paper by Ma et al., also examined the relationship between neuronal events detected through calcium recordings and resting–state hemodynamic signals through optical methods (5). They found clear evidence for the contribution of excitatory neural activity underlying resting–state hemodynamic patterns in the awake and anesthetized brain (i.e., interhemispheric/bilateral functional connectivity). Additionally, earlier work by Matsui et al. also utilizing simultaneous monitoring of neuronal calcium signals and resting–state hemodynamic signals identified two types of large–scale spontaneous patterns (i.e., global waves propagating across neocortex and transient co–activations such as interhemispheric connectivity) (6). Thus, the reviewer suggests two essential revisions. First, the present manuscript should acknowledge these prior works, discuss and compare the findings, as these prior studies are highly complementary.

We fully agree that these two studies are of high relevance and complementary to our work and now mention them prominently already in the introduction of the revised manuscript version as they confirm our findings of slow propagating calcium waves and a related hemodynamic signal.

These two studies correlated hemodynamic fluctuations with wide-field calcium imaging. Ma et al., (2016) convolved these two signals and Matsui et al., (2016) compared functional connectivity during different transient calcium signals. The results support the notion of resting-state hemodynamics being coupled to underlying patterns of excitatory neuronal activity, but particularly lower-frequency hemodynamic fluctuations were not well-predicted (Ma et al., 2016). Evidence was found that global calcium activity is linked to the hemodynamic activity and that the spatial information of the cortical network’s functional connectivity may be embedded in the phase of global calcium waves (Matsui et al., 2016).

However, we did conduct a different methodological approach: we identified calcium waves and used them as a regressor for the BOLD fMRI timecourse and asked, whether any component of the BOLD signal is related to it. Therefore, while we certainly agree with the reviewers that these two studies are important in the context of our work and are beautifully conducted, they are indeed complementary, as we start with the calcium wave, and these studies rather correlate the hemodynamic signals to the calcium activity, without defining specific events.

We thank the reviewers for pointing out these studies and now acknowledge them in the introduction and compare their findings to ours to grant the complementarity of their and our results.

Furthermore, authors should also justify the use of the current analysis technique to examine the relationship between calcium signals and rsfMRI BOLD signals and they should also discuss why they used a different approach than the approach used by prior studies as the general objective were largely similar.

As explained above, the initial idea of our study was to exclusively determine BOLD activity during the active phases of slow wave activity. As we recorded calcium slow waves of a local population with state-of-the-art methods (Stroh et al., 2013; Adelsberger et al., 2014; Zhang et al., 2017) and assured the extraction of these waves from the ongoing signal to be in line with previous studies in the field of slow wave activity (e.g. of the group of Maria Sanchez-Vives who is working on this topic since decades employing electrophysiological methods), we are confident regarding the spatiotemporal precision of our event detection. The subsequently employed event-related fMRI analysis is well suited for this approach as any type of previously defined event, may this be brief epochs of stimulus presentation as in the classical use of this analysis within an event-related design in human fMRI, or neurophysiologically defined events extracted from our calcium signal, can be used as an regressor to detect related BOLD responses. Additionally, the approach is straight-forward as it is built-in for commonly used analysis software (spm) and therefore could be of advantage for many researchers, including translational human studies which have secondary readouts from EEG-signals. Other studies relating low-frequency activity (e.g. LFP signals, or the previously mentioned large-scale calcium readouts) to rsfMRI/BOLD or hemodynamic signals in most of the cases merely correlated the ongoing signal to BOLD. At least we are not aware of any study comparing only the epochs of activity (slow wave events or “population up-states”) during slow wave activity to the BOLD signal. Our work started with this aim and later on expanded on the investigation of the ongoing signal (e.g. see our new seed-based approach) exactly to be able to compare our results to the ones of others. Nevertheless, we acknowledge the reviewers concern regarding the GLM based model and therefore now included several control analyses, which we mention in detail below.

We certainly agree with the reviewers that it is an essential point of this revision to integrate results of previous work more into the Discussion section of the manuscript as well as discussing and comparing the methods of others. We followed this request and included a new paragraph in the Introduction as well as in the Discussion section upon this topic.

Second, the comparison of BOLD signal extracted from 30 most active voxels and from an ROI (Figure 3—figure supplement 1) showed apparent delay in the peak and onset. Is this in anyway related to certain propagation characteristics as alluded by (6) and prior work done by the authors (7)? If so, the authors should characterize this phenomenon as it is highly interesting.

We definitely agree with the notion of this comment, that indeed our global activation may represent traveling waves as it is suggested by the work the reviewers mention but also by many others (e.g. Massimini et al., 2004). Nevertheless, to draw the conclusion regarding propagation characteristics upon a delay in the peak and onset of the BOLD signal would be an overinterpretation of our data. With the slow sampling rate of fMRI (1s) we are undersampled to be able to make this claim. Again, we agree that it is very likely that this propagation exists, as it is exactly what the work of Matsui et al. and previous work from our lab (Stroh et al., 2013) shows, however both using methods with high sampling rates, but the inherent slow sampling rate of classical fMRI will unfortunately not be able to resolve the propagation speed in the rodent brain (~ 30 mm/s). Recently introduced so called line-scanning techniques (Yu et al., 2014) would be a way to resolve traveling slow waves in fMRI data in future studies as we now mention in the discussion of the revised manuscript version.

In addition, a recent work by Leong et al. also demonstrated that the low frequency or slow waves can propagate robustly through long–range excitatory projections. Offering a multi–faceted view in the context of rsfMRI will definitely increase the impact of the present study.

We agree with the reviewer’s comment and expanded the discussion upon this topic to ensure the mentioned multi-faceted view in the context on rsfMRI and cite the work of Leong et al., 2016.

5) At present, the authors attempt to claim that cortex–wide BOLD activity was identified only in slow wave state but not in the active desynchronized state and attribute their findings to different brain states by conducting the experiments under isoflurane anesthesia and medetomidine sedation. As eluded already above, this claim is generally too broad and simplistic. Moreover, the present finding is insufficient to support such assertions, unless awake animal experiments are conducted. In fact, this message distracts the reader from the core findings of the study, which is correlating a neuronal event (i.e., calcium slow wave activity) to brain–wide rsfMRI hemodynamic signals. The core message however, is not examining such relationship at different "states", which was the major reason the paper was rejected in the first place. The authors are certainly aware that brain states fluctuate temporally whereby different states are only dominant for a certain period. Moreover, certain features of the rsfMRI such as interhemispheric functional connectivity are also present at both states. Hence, it is a gross oversimplification to assign binary representations for the characterized brain states. Scaling back on these claims and better emphasizing the strength of the paper – i.e. relationship between the calcium waves and the fMRI will address the reviewers' concerns.

We agree with the reviewers that the emphasis of the manuscript – indeed the relationship between slow calcium waves and BOLD fMRI – has to regain strength in this work. We are definitely aware of the fact that brain states may fluctuate over time and represent ongoing phenomena, which are not binary and we never intended to claim any categorical or binary state characteristics. We agree that this would be an oversimplification of the data presented. We apologize for giving this impression in the previous version of the manuscript and address the reviewers concerns by scaling back on the brain state related claims and by emphasizing the strength of the manuscript as mentioned above, focusing on the relation of slow waves and their corresponding BOLD correlates. Just to clarify this issue, we want to state that the presented types of neuronal activity (slow wave and persistent activity), represent two extreme cases of continuously fluctuating states of the neural network. Classifications of such states indeed have been done before (see below figure of Renart et al., for in vivo data and Mattia and Sanchez-Vives for simulated networks), but as the reviewers correctly point out, are neither the emphasis of our study nor the correct envision for our data. Exactly because of the fact that certainly, the underlying anatomical connectivity is identical in both conditions, overlapping activity signatures likely do exist. Indeed we now can demonstrate that this is the case: when inspecting the complete component set revealed by group ICA of the slow wave data, also here classical rs-networks and interhemispheric connectivity can be identified. For example the auditory resting state network is present during both types of activity (Figure 5—figure supplement 1 and Figure 5—figure supplement 4).

So the reviewers are indeed right, although apparently dominated by the cortex-wide BOLD activation related to slow wave activity, also rs-networks (which can be found under different conditions) remain to be present during slow wave activity – a finding which seems reasonable as these networks are discussed to mirror basic functional connectivity.

We nonetheless base our concept on two conditions in which a) large amplitude calcium waves mirroring slow oscillations are present (slow wave activity) and b) a sedated condition, in which these slow oscillations are absent (persistent activity). Thereby we are not contradicting or challenging earlier studies, we rather provide an additional variable, which is already used in basic neuroscience since many years, and translate the concept of relating local neurophysiological measures of slow waves to the field of neuroimaging. We however, feel that investigating the conditions under which no slow waves are present highlights the specificity of our findings and serves as an important control condition. Nevertheless, as this control is not the main focus of our study, we now present exclusively the slow wave activity related results in the main figures (please also refer to Fig. S1 from Renart et al., 2010, and Figure 4 from Mattia and Sanchez-Vives, 2012).

6) The authors stated that slow wave activity represents a brain state indispensable for memory consolidation. Knowing that the hippocampus plays a major role in memory consolidation, it would be interesting to also explore the BOLD fMRI signals in the hippocampus, but this is not an essential revision.

It was previously reported that slow waves also engage the hippocampus (Sirota, Csicsvari, Buhl, & Buzsaki, 2003; Hahn, Sakmann and Mehta, 2006; Ji and Wilson, 2007; Busche et al., 2015), suggesting that the slow wave associated excitation plays a synchronizing role in cortico-hippocampal functional coupling (Hahn, Sakmann, & Mehta, 2006).

A very recent study by Chan et al., (2017) which we now cite, investigates this issue in beautiful detail.

As indeed we occasionally observed hippocampal BOLD activation during slow waves which we already reported in the previous version in the manuscript, we thank the reviewers for this comment and now conducted an atlas-based seed-ROI analysis for the hippocampus (Figure 6—figure supplement 2). This analysis revealed that voxel in the hippocampal ROI correlate well with cortical voxel, again spanning the entire cortical surface during ongoing slow wave activity, but lacking such a pan-cortical component under medetomine sedation.

7) As stated in the previous review there was serious concerns related to the computational/analytic processing of the fMRI signal.– The authors should provide a convincing argument that their "global signal" is not a processing or acquisition artifact.

We fully agree with the reviewer´s comment that we have to rule out any sources which may result in the observed cortex-wide activity patters, and which are not related to the optically recorded slow waves. We now conducted a new set of analyses (set in italics) and we strongly believe that we demonstrate this very relation beyond reasonable doubt. We here mention possible sources of artefacts and address them point by point:

a) Technical artefact

When slow wave vectors used as regressors for the subsequent BOLD fMRI analysis are temporally distorted, either by swapping them between animals/experiments or by temporally mirroring them, pan-cortical activation vanishes completely (Figure 3—figure supplement 4), *and the correlation between the GLM timecourse and the ICA component disappears (Figure 5)*. These results speak for the slow wave events being temporally informative for these analyses.

b) Physiological/non-physiological noise artefact

See A); in addition we compare these results to a type of neural activity which lacks the slow wave component and which is phenomenologically similar to an persistent/active type of ensemble activity when observing the calcium readouts (Figure 5—figure supplement 3). This physiological control for slow wave specificity of our results *indeed revealed the absence of a pan-cortical component also in seed-based analyses.* As both network conditions were achieved in the same animal, this rules out potential confounds or artifacts related to subject-specific noise, which would have been present in both measurements.

c) Data processing artefact

We rely on three independent methods for the analysis of our data: (1) a classical GLM-based approach (event-related analysis employing simultaneously acquired slow calcium wave events; Figure 2–Figure 4), and two model-free techniques (2) ICA (Figure 5) and (3) seed-based analysis (Figure 6). We identify pan-cortical activity in all of these analyses. While each method clearly has its limitations, and we significantly expanded on the IC analysis (see below), we believe that it is highly unlikely that an analysis artefact is the source of our finding. Even more so, as pretreatment (preprocessing) and handling of the data in the analysis types we used is slightly different (see Material and Methods part) it is unlikely that a data processing artifact is causing the global signal in the slow wave related data only. Also, we now show that regressing out the slow wave vector extracted from the calcium measurements from the data leads to the absence of pan-cortical components in the IC analysis (Figure 5—figure supplement 2).

d) Artefact of non-neuronal source.

We employed calcium indicators reflecting only neuronal activity (GCaMP6 under control of an excitatory neuronal promoter) and could identify the same slow calcium wave with the same dynamics, ruling out major non-neuronal contributions.

ICA decomposition always produces a substantial number of noise components from white matter, CSF, cardiac and respiratory sources. The authors do not mention if or how they dealt with them. They do not mention if their data was pre–whitened prior to submission to ICA. They do not present their component set and discuss it. Further, claims regarding the 'event–related' regressor and global signal would also have been burnished by thoughtful treatment of other potential extraneous signal noise such as drift, fMRI lag, temporal autocorrelation etc.

In the previous version of the manuscript we initially extracted 17-22 components for each experiment, based on the minimum description length criterion (MDL), which should avoid choosing too few components (please see also below). In this previous analysis we calculated the number of components for each dataset separately employing MDL. We now implemented a group ICA approach (equally based on the MDL method for component identification) where we show the complete component set and identify meaningful components by comparing the network activity they reveal with the work of others. Based on the results of the group ICA we chose the number of components for the single subject IC analysis.

Further evidence on the specificity of the pan-cortical component (other than its absence in persistent state, see above) is also presented now by regressing out the timecourse derived by the slow wave vector obtained from the calcium signal, leading to a complete absence of this pan-cortical component.

We now added detailed information about the ICA processing steps (as well as about the GLM) in the Material and methods section. The ICA data is pre-whitened and drift corrected, the GLM data is preprocessed in line with other studies (realignment, reslicing and smoothing), serial autocorrelation was performed and we now describe these steps accordingly in the revised manuscript. We apologize that this information was insufficiently provided in the previous version.

Why has this global signal not been described in rsfMRI before, and if so, please cite the literature where such a "global signal" has been described before.

A pan-cortical signal component had previously been described in other studies (see below), but the origin of this signal was not neurophysiologically well-defined. But since the implementation of calcium recordings alongside fMRI recordings, the same pan-cortical activation, following our event-related approach, had been reported in mouse by the group of Markus Rudin (Schlegel et al., 2017).

As explained above, we would not necessarily relate slow waves induced by deep anesthesia to the classical definition of resting state activity as this has been defined in awake human subjects. Nevertheless, as mentioned, the extent of cortical recruitment by a locally measured slow wave event (our event-related analysis), as well as the networks present during ongoing slow wave activity (our seed-based and ICA approach), is of high interest for the community as slow wave activity is an extensively studied phenomenon.

Indeed, under similar conditions such global signals have been described, e.g. by Liu et al., 2011 as well as by Kalthoff et al., 2013. As already speculated by Kalthoff and colleagues, the global signal removal step – often applied in rs studies – might conceal this very component identified here, under experimental conditions in which slow waves could in principle occur (rather deep anesthesia). Kalthoff et al., (2013) speculated that such a signal exists and may be related to slow wave activity.

Also, Murphy and colleagues described a similar result for EEG activity under Propofol anesthesia in human subjects (Murphy et al., 2011; see their Figure 1). Deco et al., (2014) modeled that when neuromodulators decrease to very low levels (as it is likely the case during deep anesthesia), slow waves become global and resting-state networks merge into a single undifferentiated, synchronized network.

In their beautifully designed study employing optogenetics, Leong et al., (2016) also show a broad cortical activation for waves traveling upon stimulation at a low frequency (1Hz; see their Figure 2 upper panel).

Also, Matsui et al. show global cortical waves (compare their Figure 2 upper last panel).

But clearly, the concept of relating only the slow wave events to the simultaneously acquired BOLD fMRI signal is novel and represents the main advance of this study. We now cite the literature mentioned here in the revised version of the manuscript.

– It is difficult to understand why the rsfMRI ICA model suggests that under deep anesthesia, a neuronal component appears that represents wide activation. The rsfMRI ICA is not contingent on the calcium wave model – it was just regressed against it. Please explain this confusion.

As explained it is likely that the global activity we observe is actually a traveling/expanding wave which is passed on to neighboring local networks (Sanchez-Vives, Massimini and Mattia, 2017). The work of Matsui et al. using similar readouts (large-scale calcium imaging) also shows a widespread activation during slow wave activity. Additionally, the merging of activity to a global signal was modeled in silico by Deco et al., (2014). For details please see also replies above.

– Please explain how your data unambiguously demonstrates that the fMRI signals come from neuronal sources.

For the exclusion of *non-neural (or non-physiological)* source please see above points A-C. Specifically for the exclusion of *non-neuronal* sources:

D: We employed calcium indicators reflecting only neuronal activity (GCaMP6 under control of an excitatory neuronal promoter) and could identify the same slow calcium wave with the same dynamics, ruling out major non-neuronal contributions.

Additionally, to show what removing the influence of the neuronal source produces in the BOLD signal, we now regressed out the calcium wave regressor from our data, which leads to the complete elimination of meaningful results in the BOLD signal when used in the ICA.

– Please provide more methodological detail for the fMRI methods used.

We extended the method section accordingly and also structured the fMRI methods by analysis approaches by using sub-headings.

– The pipeline used (HRF, task–based contrast modeling) is very vulnerable to artifacts. The link with neuronal activity is novel, and thus, it will be extremely important to be very cautious with making claims. This should be specifically addressed in the discussion.

We scaled back on our claims and addressed these issues in the discussion. We put our results, which fit very well to the assumptions previous authors made, in perspective to the literature cited above.

– The authors should clarify whether they performed group or single subject ICA notwithstanding their presentation of single subject results.

In the revised version of the manuscript we now performed group ICA (Figure 5—figure supplement 1). We additionally present single subject results. We extended the Materials and methods section accordingly and also clarify this in the results and apologize for the confusion.

– While the authors in their rebuttal provide some additional information as to their conduct of the ICA, this was not reassuring. At the heart of every ICA is the choice the researcher must make as to how many components to model. Model order is critical to the number and kind of components produced and the interpretation of results. Contrary to the authors assertion, the choice of model order is not "arbitrary". Please clarify were 4 or 17 or 22 PCs/ICs specified in the model? Were the number of PCs and ICs the same or different? Were different ICs used in different animals? If only 4 ICs were specified, one wonders if this drove the production of a single, ostensibly 'global' signal. If 17–22 ICs were specified, one wonders what all those other ICs were.

In our new group ICA (Figure 5—figure supplement 1) we now defined the number of components by using the minimum description length criterion (MDL; Li et al., 2007; see also below). This statistical parameter estimation revealed 37 ICs, which were then inspected visually, and ICs that showed activity outside of the brain only were discarded. By this procedure a number of 23 ICs remained, which in our view provides a good equilibrium between extracting sufficient components for not artificially producing a single global signal – as mentioned by the reviewer – and identifying a reasonable number of meaningful components in terms of resulting network activity (see our results explained above). Based on the result of this group IC analysis we set the number of components for the ICAs performed on single-subject basis to 20 (Figure 5, Figure 5—figure supplement 1 and Figure 5—figure supplement 4), as group analysis provides a good basis to choose components for each individual subject and 20 is a reasonable starting number for ICs in single subject analysis within the framework of resting state data (e.g. Li and Wang, 2015).

We apologize for the previous confusion regarding PCs/ICs: PCs are used for pre-whitening of the data. For the group-level-ICA (n=4), first, the number of principal components (PC) is calculated from all data, taking into account the number of independent components. In our case this resulted in 56 PCs for the pre-whitening step. For ICAs performed on single subject basis (n=15) the number of PCs used for pre-whitening equals the number of ICs, as here only one calculation step is performed.

We rephrased the Materials and methods section accordingly to make our approach better comprehensible. We explain how numbers of ICs were derived, state the number of PCs in the pre-whitening steps and provide all the parameters used within the employed “Group ICA of fMRI Toolbox (GIFT)”-toolbox.

–The authors should discuss the significant drawbacks of ICA (as opposed to e.g. cluster or seed based methods) as it produces a large number of noise components along with actual gray matter maps. There is an entire field of research dedicated to the better, automated identification of components. The authors have not detailed any of their QC process, presented their component sets or even acknowledged the context and complexity of component identification in ICA. The authors need to discuss the issues associated with this approach.

We thank the reviewer for this comment; we expanded the method section and the discussion accordingly. ICA is only one of several analyses we performed and our finding of pan-cortical BOLD activity related to slow wave activity has been identified by different paralleled approaches, now additionally by seed-based methods. Importantly, here the main signal intensities in the seed-ROIs (S1 and hippocampus) correlate high with the normalized signal of the pan-cortical ICs in those animals (Figure 6 and Figure 6—source data 1 – Table 2 for S1 data; Figure 6—figure supplement 2 and Figure 6—source data 2 – Table 3 for hippocampus data). We are aware of the drawbacks of ICA and for clarification now show the complete component set, including the potential noise components. For identification of components we used the minimum description length (MDL) criterion, which was specifically developed for ICA applied to fMRI data (Li et al., 2007) and is implemented in the GIFT toolbox we used. We discuss the drawbacks of ICA now thoroughly in the Discussion section.